# Towards Neural Scaling Laws for Foundation Models on Temporal Graphs

## Abstract

Temporal graph learning aims to extract knowledge from dynamic network data to predict future interactions. The key question is, given a set of observed temporal graphs, is it possible to forecast the evolution of an unobserved network within the same domain? To answer this question, we present Temporal Graph Scaling (TGS) dataset, a large collection of temporal graphs consisting of *eighty-four* ERC20 token transaction networks collected from 2017 to 2023. Next, we assess the transferability of Temporal Graph Neural Networks (TGNNs) in temporal graph property prediction by pre-training on up to 64 token transaction networks and evaluating their downstream performance on *twenty* unseen token networks.

We observe that the neural scaling law, previously identified in NLP and computer vision, also holds in temporal graph learning. Specifically, pre-training on a larger number of networks results in enhanced downstream performance. To the best of our knowledge, this study is the first empirical demonstration of transferability to unseen networks in temporal graph learning. Notably, on *thirteen out of twenty* unseen test networks, our largest pre-trained model using zero-shot inference can outperform fine-tuned TGNNs on each test network. We believe that this work is a promising first step towards developing foundation models for temporal graphs. The implementation of Temporal Graph Scaling can be accessed at https://anonymous.4open.science/r/ScalingTGNs.

## 1 Introduction

Foundation models have revolutionized various fields such as natural language processing (NLP) Bubeck et al. (2023); Brown et al. (2020); Rasul et al. (2024) and computer vision (CV) Radford et al. (2021); Awais et al. (2023) by providing robust pre-trained architectures that can be transferred to a multitude of tasks. Foundation models aim to learn from large amounts of pre-training data and transfer the knowledge to downstream unseen tasks. These models have been recognized for their remarkable transfer capabilities and promising efficacy with few-shot and zero-shot learning on novel datasets and tasks Bommasani et al. (2021); Dong et al. (2023); Rasul et al. (2024). Despite advances in NLP and CV, foundation models in graph representation learning remain relatively unexplored. For example, there has been some notable work on foundational models for graph neural networks (GNNs) that demonstrate the potential of these models Mao et al. (2024); Galkin et al. (2023); Beaini et al. (2023); Méndez-Lucio et al. (2022). However, most research has focused on static graph learning, leaving the exploration of temporal graph neural networks largely untapped.

To effectively train foundation models, a large collection of datasets is essential. Networks within the same domain often exhibit similar trends and statistics Jin & Zafarani (2020). These datasets are crucial for assessing the performance of TGNNs, driving innovation, and ensuring that new methods can be generalized across various applications. To facilitate research on foundation models for temporal graphs, we introduce the Temporal Graph Scaling (TGS) benchmark, a comprehensive dataset containing 84 novel temporal graphs derived from Ethereum transaction networks. TGS provides temporal networks with up to 128K nodes and 0.5M edges, totaling 3M nodes and 19M edges across all networks. These datasets also vary in their time duration and helps facilitate the training of foundation models for temporal graph learning.

Quick adaptation of a foundation model to novel unseen data is crucial, especially in financial token networks, where new datasets frequently emerge and the costs of training multiple models become

prohibitive Shamsi et al. (2022); Zhang et al. (2023). To achieve this, we must first study how transferrable a pre-trained temporal graph model is to unseen networks. Therefore, we propose the first algorithm for pre-training TGNNs on multiple temporal graphs, called the TGS-train algorithm. Models that are trained on multiple networks are then referred to as multi-network models. With only zero-shot inference, our multi-network models achieve significant performance advantages over models trained on individual test networks. This demonstrates the high potential of transferability of large pre-trained models on temporal graphs. We also demonstrate that training on a larger number of temporal graphs results in stronger downstream performance. Figure 1 shows the scaling behavior of our multi-network model. The average performance of the multi-network model on twenty unseen token networks increases as the number of networks used for training increases.

Our main contributions are as follows:

- **Novel Collection of Temporal Networks.** We release a comprehensive collection of $84$ labeled datasets derived from token transaction networks for the graph property prediction task. These datasets provide the foundation for studying scaling behavior, transferability and multi-network learning on temporal graphs.

- **First Multi-network Training Algorithm for Temporal Graphs.** To the best of our knowledge, we propose the first training algorithm, named TGS-train, that enables TGNNs to train on multiple networks at once.

- **Neural Scaling Law on TGNNs.** We explore the potential of foundation models on temporal graphs by showing that neural scaling law also applies to temporal graphs: training TGNNs with more temporal graphs (up to $64$) offers a significant performance boost in downstream test networks.

- **Transferability Across Networks.** We demonstrate that by pre-training on a large number of temporal graphs, our multi-network model is directly transferable to $20$ downstream unseen token networks while outperforming single models trained on the test networks. This shows that it is possible to learn an overall distribution across temporal graphs and transfer it to novel networks.

**Reproducibility.** Our code is available on 4open.science. The TGS datasets are publicly available on Dropbox (during the anonymity period).

## 2 RELATED WORK

**Temporal Graph Benchmarks.** Numerous graph benchmark datasets have been introduced to advance research within the temporal graph learning community. Poursafaei et al. (2022) introduced six dynamic graph datasets while proposing visualization techniques and novel negative edge sampling strategies to facilitate link prediction tasks of dynamic graphs. Following the good practice from OGB Hu et al. (2020), Huang et al. (2023) introduced TGB, which provides automated and reproducible results with a novel standardized evaluation pipeline for both link and node property prediction tasks. However, these datasets belong to different domains, making them unsuitable for studying the scaling laws of neural network models trained with a large number of datasets from the same domain. Li et al. (2024) provide a temporal benchmark for evaluating graph neural networks in link prediction tasks, though their focus does not extend to training on multiple networks. Conversely, the

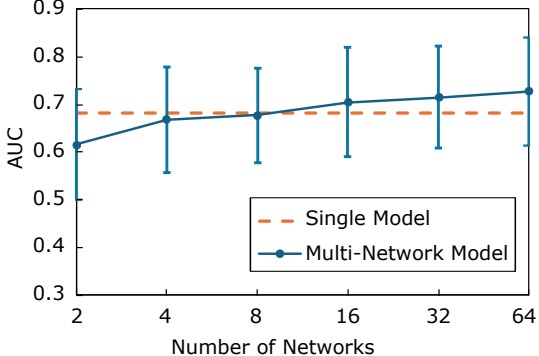

Figure 1: **Scaling behavior of multi-network models.** The performance of multi-network trained on $2^n$ where $n \in [1, 6]$ compared to a *single model* that is trained on each test dataset and a simple baseline such as *persistence forecast*.

Live Graph Lab dataset by Zhang et al. (2023) offers a temporal dataset and benchmark, employed for tasks like temporal node classification using TGNNs. This work aims to explore multi-network

training and understand the transferability across temporal graphs. Therefore, we curate a collection of temporal graphs rather than focusing on individual ones as in prior work.

**Discrete Time Dynamic Graphs.** A common approach in discrete time models treats each snapshot individually and captures spatial characteristics, then adopts an RNN-based method to learn temporal dependencies Seo et al. (2016); Sankar et al. (2019); Chen et al. (2022); Li et al. (2019); Shamsi et al. (2024). GCRN stacks a graph CNN for feature extraction and an LSTM cell for temporal reasoning Seo et al. (2016). Differentiating from GCRN, EvolveGCN Pareja et al. (2020) uses RNN to control the parameters of a GCN at each snapshot. Employing two attention blocks, DySat first generates static node embeddings at each snapshot by running a GAT style GNN, and then computes new embeddings using a temporal self-attention block Sankar et al. (2019). In the most recent work, GraphPulse Shamsi et al. (2024) leverages Mapper, a key tool in topological data analysis, to extract essential information from temporal graphs. However, in all these studies, the training process of every model was limited to a single dataset, and the effectiveness of training TGNs with diverse networks to enhance their generalization capabilities is unexplored.

**Neural Scaling Laws.** Neural scaling laws characterize the relationship between model performance and three main factors: number of parameters, size of training datasets and amount of computation Rosenfeld et al. (2020); Kaplan et al. (2020); Abnar et al. (2022). These relationships are usually described as a power law, which can be understood by observing learning as a movement on a smooth data manifold Bahri et al. (2021). Bahri et al. (2021) exhibited all four scaling regimes with respect to the number of model parameters as well as the dataset size, underscoring different mechanisms driving improvement in loss. The authors provided valuable insights into the design and training of mixed-model generative models by studying mixed-modal scaling laws, indicating the generality of scaling laws across different domains and applications. Recently, Liu et al. (2024) investigated neural scaling laws for static graphs by observing the performance of GNNs given increases in the model's size, defined by the number of layers and parameters, and training set size, defined by the number of edges. To the best of our knowledge, we are the first to investigate neural scaling laws for temporal graphs.

**Foundation Models**. The foundation model is an emerging paradigm that aims to develop models capable of generalization across different domains and tasks using the knowledge obtained from massive data in the pre-trained stage. Recently, Rasul et al. (2024) introduced Lag-Llama, a general-purpose foundation model for univariate probabilistic time series forecasting based on a simple decoder-only transformer architecture that uses lags as covariates. Galkin et al. (2023) introduced ULTRA, a foundation model for knowledge graphs, which handles complex relational data and supports diverse downstream tasks effectively. Similarly, Beaini et al. (2023) presented Graphium, a collection of molecule graph datasets that facilitate the development of foundation models for molecular applications, highlighting the importance of domain-specific datasets in enhancing the performance and generalizability of foundation models. Lastly, Xia et al. (2024) proposed OpenGraph, an initiative towards open foundation models for graphs, emphasizing the need for transparency, reproducibility, and community-driven advancements in graph representation learning. These works underscore the growing recognition of the importance of foundation models and their transformative potential across various domains, such as molecular graphs. However, foundation models for temporal graphs remain unexplored.

## 3 PRELIMINARIES

Temporal Graphs are generally categorized into two types: Continous Time Dynamic Graphs (CTDGs) and Discrete Time Dynamic Graphs (DTDGs) Kazemi et al. (2020). We focus on DTDGs because this approach aligns well with our objective of capturing and analyzing the graph's dynamics at specific time intervals, such as on a weekly basis. In DTDGs, the graph's temporal evolution is represented in discrete time steps, simplifying the analysis and modeling of large-scale temporal multi networks. Each time step provides a snapshot of the graph at a specific moment, facilitating straightforward comparisons and the identification of temporal patterns.

**Definition 1** (Discrete Time Dynamic Graphs)**.** *DTDGs represent the network as a sequence of graph snapshots denoted as $\mathcal{G} = \{\mathcal{G}_{t_1}, \mathcal{G}_{t_2}, \mathcal{G}_{t_3}, \ldots, \mathcal{G}_{t_n}\}$ where $t_i < t_j$. Each $\mathcal{G}_{t_i} = (\mathcal{V}_{t_i}, \mathcal{E}_{t_i}, \mathbf{X}_{t_i}, \mathbf{Y}_{t_i})$ is the graph at timestamp $t_i$, where $\mathcal{V}_{t_i}$ and $\mathcal{E}_{t_i}$ represent the set of nodes and edges, $\mathbf{X}_{t_i}$ denotes the node feature matrix, and $\mathbf{Y}_{t_i}$ represents the edge feature matrix in graph $\mathcal{G}_{t_i}$. Therefore, a collection*

*of discrete-time dynamic graphs is defined as $D = \{\mathcal{G}^1, \mathcal{G}^2, \ldots, \mathcal{G}^m\}$, where $m$ is the number of DTDGs.*

**Temporal Graph Property Prediction.** For the task of temporal graph property prediction, we aim to forecast a temporal graph property within a future time interval in a DTDG. More specifically, given a DTDG $\mathcal{G}$, we consider a time duration $[t_{\delta_1}, t_{\delta_2}]$, where $\delta_1$ and $\delta_2$ are non-negative integers with $\delta_1 \leq \delta_2$. Then at a specific time $t_k$, the goal is to predict the target graph property within the specified future interval $[t_{k+\delta_1}, t_{k+\delta_2}]$. Further details about our task formulation, including the definition of our graph property prediction and example of other property prediction tasks on graphs, are provided in Appendix Section C. .

**Hyperbolic Graph Neural Networks.** Hyperbolic geometry has been increasingly recognized for its ability to achieve state-of-the-art performance in several static graph embedding tasks Yang et al. (2021). HTGN is a recent hyperbolic work that shows strong performance in learning over dynamic graphs in a DTDG manner. The model employs a hyperbolic graph neural network (HGNN) to learn the topological dependencies of the nodes and a hyperbolic-gated recurrent unit (HGRU) to capture the temporal dependencies. Temporal contextual attention (HTA) is also used To prevent recurrent neural networks from only emphasizing the most nearby time and to ensure stability along with generalization of the embedding. In addition, HTGN enables updating the model's state at the test time to incorporate new information, which makes it a good candidate for learning the scaling law of TGNNs. In our TGS framework, we use the HTGN architecture as part of our multi-network model because it excels in dynamic graph learning through hyperbolic geometry. Its strong performance makes it a valuable addition to our approach. We further describe the HTGN in Appendix Section D.

# 4 DATASET

We utilize a dataset of temporal graphs sourced from the Ethereum blockchain Wood et al. (2014). In this section, we will describe Ethereum, explain our data pipeline, and conclude by defining the characteristics of the resulting dataset.

**Ethereum and ERC20 Token Networks.** We create our transaction network data by first installing an Ethereum node and accessing the P2P network by using the Ethereum client Geth (`https://github.com/ethereum/go-ethereum`). Then, we use Etherum-ETL(`https://github.com/blockchain-etl/ethereum-etl`) to parse all ERC20 tokens and extract asset transactions. We extracted more than sixty thousand ERC20 tokens from the entire history of the Ethereum blockchain. However, during the lifespans of most token networks, there are interim periods without any transactions. Additionally, a significant number of tokens live for only a short time span. To avoid training data quality challenges, we use 84 token networks with at least one transaction every day during their lifespan and are large enough to be used as a benchmark dataset for multi-network model training.

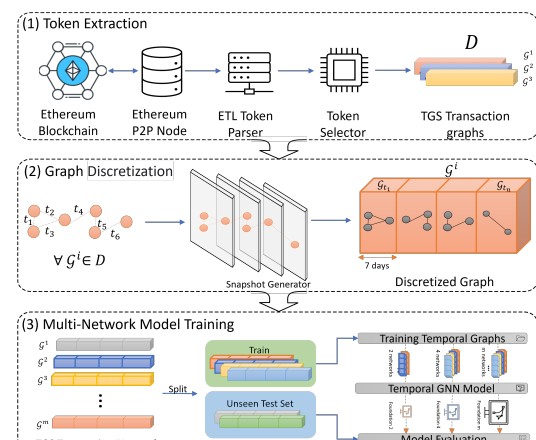

Figure 2: **TGS overview.** (1) *Token extraction*: extracting the token transaction network from the Ethereum node. (2) *Discretization*: creating weekly snapshots to form discrete time dynamic graphs. (3) *Multi-Network Model Training*: TGS transaction networks are divided randomly into train and test sets. We train the MNs on a collection of training networks. Lastly, MNs are tested on 20 unseen test networks.

**Temporal Networks.** Each token network represents a distinct temporal graph, reflecting the time-stamped nature of its transactions. In these networks, nodes (addresses), edges (transactions), and edge weights (transaction values) evolve over time, capturing the dynamic behavior of the network. Additionally, these networks differ in their start dates and durations, introducing further variation in their evolution. While each token network operates independently with its own set of investors, they exhibit common patterns and behaviors characteristic of transaction networks. These similarities

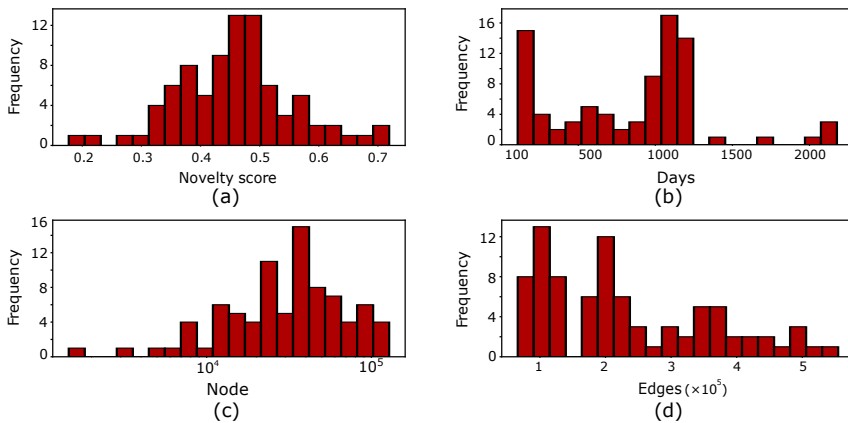

Figure 3: Network statistics of TGS networks: (a) Novelty score, (b) number of days, (c) number of nodes, and (d) number of edges.

allow the model to learn and generalize from these patterns across different networks. Collecting temporal graphs from different ERC20 token networks allows for comparative analysis, uncovering in-common patterns and unique behaviors. This strengthens the model's ability to generalize and improves its robustness.

Figure 2 illustrates the TGS overview from dataset extraction to the multi-network (MN) model training step.

**Dataset Statistics.** Our TGS dataset is a collection of $84$ ERC20 token networks derived from Ethereum from 2017 to 2023. Each token network is represented as a dynamic graph, in which each address and transaction between addresses are a node and directed edge, respectively. The biggest TGS token network contains $128,159$ unique addresses and $554,705$ transactions, while the smallest token network has $1,454$ nodes. TGS contains a diversity of dynamic graphs in terms of nodes, edges and timestamps, which are shown in Figure 3. Details on statistics are given in Appendix A. The figure shows that most networks have more than 10k nodes and over 100k edges. The lifespan of TGS networks varies from $107$ days to 6 years, and there exists at least one transaction each day. Figure 3.a shows the novelty scores, i.e., the average ratio of unseen edges in each timestamp, introduced by Poursafaei et al. (2022). Figure 3 shows that most of the $84$ networks have novelty scores greater than $0.3$, indicating that each day sees a considerable proportion of new edges in these token networks. We adopt a $70 - 15 - 15$ split of train-test-validation for each token network and calculate the surprise score Poursafaei et al. (2022), which indicates the number of edges that appear only in the test data. As Table 4 shows, the token networks have quite high surprise values with an average of $0.82$. We also provide the node, edge and length distribution for train and test sets separately in Figure 6. Overall, train set datasets mostly have more nodes than those in the test set, while the number of edges and days are in the same range for both. A detailed overview of the characteristics of the TGS datasets is presented in Appendix A.

## 5 METHODOLOGY

We use Temporal Graph Neural Networks (TGNNs) as the multi-network model architecture. We choose the state-of-the-art Hyperbolic Temporal Graph Network (HTGN) Yang et al. (2021) as an example architecture for experiments. This section explains our choice and details our training algorithm on multiple networks.

### 5.1 MULTI-NETWORK TRAINING ON TEMPORAL GRAPHS

Existing temporal graph learning models typically train on a single temporal graph, limiting their ability to capture similar behaviors and generalize across different networks Rossi et al. (2020); Yang et al. (2021). We introduce TGS-train, the pioneering algorithm designed to train across multiple temporal graphs by modifying a state-of-the-art single network training model with two crucial

---

**Algorithm 1:** TGS-train: Multi-Network Training for Temporal Graphs

---

**Input:** A Temporal Graph Dataset $D = \{\mathcal{G}^1, \mathcal{G}^2, \ldots, \mathcal{G}^m\}$, where $\mathcal{G}^i = \{\mathcal{G}^i_{t_1}, \mathcal{G}^i_{t_2}, \ldots, \mathcal{G}^i_{t_n}\}$
$m$ = Number of networks in training, **TGNN** and **Decoder**
**for** *each epoch* **do**
    Shuffled ($D$) // IID training
    **for** *each network $\mathcal{G}^i \in D$* **do**
        Initialize historical embeddings (reset) // context switching
        **for** *each training snapshot $\mathcal{G}^i_{t_j} \in \mathcal{G}^i$* **do**
            $\mathcal{H}_{t_i} = \mathbf{TGNN}(\mathcal{G}^i_{t_j})$
            $\hat{y}_{t_i} = \mathbf{Decoder}(\mathcal{H}_{t_j})$
            $\mathcal{L} = \mathbf{Loss}(y_{t_i}, \hat{y}_{t_j})$
            Backpropagation
            Update historical embeddings with $\mathcal{H}_{t_j}$
        Evaluate on the validation snapshots of $\mathcal{G}^i$
    Average validation results across all datasets to select the best model
    Save the best model for inference

---

steps: *shuffling* and *resets*. These steps, as we describe below, render the algorithm network-agnostic, capable of learning from various temporal graphs to generalize effectively to unseen networks.

Algorithm 1 shows TGS-train in detail. As the first step, we load a list of $m$ temporal graphs $D = \{\mathcal{G}^1, \mathcal{G}^2, \ldots, \mathcal{G}^m\}$, where each temporal graph $\mathcal{G}^i$ is represented as a sequence of snapshot $\{\mathcal{G}^i_{t_1}, \mathcal{G}^i_{t_2}, \ldots, \mathcal{G}^i_{t_n}\}$. For each epoch, we shuffle the orders of the list of datasets $D$ to preserve the Independent and Identically Distributed (IID) assumption of neural network training.

**IID training.** To preserve the IID assumption in neural network training, we include a shuffling step at each epoch. The randomized ordering of networks during training at each epoch is important because it helps prevent the model from learning spurious correlations that could arise if the data were presented in a fixed order. By shuffling the datasets, we promote randomness in the training process, which contributes to more robust and generalizable model performance. Sequentially, for each dataset $\mathcal{G}^i$, we first initialize the historical embeddings, then train the model end to end (i.e., encoder-decoder) on each dataset $\mathcal{G}^i$ in a similar manner of training a single model, and evaluate the performance on the corresponding validation set of dataset $\mathcal{G}^i$. After training on $m$ datasets from $D$, we compute the average validation results across these datasets. This average is used to select the best model, which is then saved for inference. Early stopping is applied if needed.

**Context switching.** Many TGNNs store and utilize node embeddings from previous timestamps at later timestamps; we refer to those embeddings as *historical embeddings* Yang et al. (2021); Chen et al. (2022); Pareja et al. (2020). Resetting historical embeddings at the beginning of each epoch is a key step in training a temporal model across multiple networks for several reasons. First, it helps prevent the model from carrying over biases or assumptions from one network to another, ensuring that it can adapt effectively to the unique characteristics of each network. Starting with fresh historical embeddings at the beginning of each epoch enables the models to learn the most relevant and up-to-date information from the current network, improving performance and generalization across different networks. Additionally, resetting historical embeddings can help mitigate the issue of *catastrophic forgetting*, where the model may gradually lose information about previous networks as it learns new ones.

**Time complexity analysis.** The TGS-train algorithm has the same complexity as training the single model across all the training networks. Specifically, the time complexity for HTGN using the TGS-train algorithm is $O(m \cdot (N_{max}dd' + d'|\mathcal{E}_{max}|))$ where $m$ is the number of training networks, $N_{max}$ is set to the maximum number of nodes of networks in the training set, $d$ and $d'$ are the dimensions of the input and output features while $|\mathcal{E}_{max}|$ is the maximum number of edges in a snapshot.

**Inference on an unseen network.** To evaluate the transferability of each multi-network model, we test the model on unseen datasets. To obtain testing data, we divide TGS into two disjoint sets, where one set is used for training obtained by randomly selecting 64 token networks, and the remaining

20 token networks are used to evaluate the performance. We begin by loading all the weights of multi-network models, including the pre-trained encoder and decoder parameters, while initializing fresh historical embeddings. Then, we perform a single forward pass over the train and validation split to adapt the historical embeddings specific to the testing dataset.

# 6 EXPERIMENTS

Weekly forecasts are common in the financial context for facilitating financial decisions Kim et al. (2021). Similarly, for the temporal graph property prediction task (defined in Section 3), we set $\delta_1 = 3$ and $\delta_2 = 10$, thus predicting the graph property over weekly snapshots. Experimentally, we use the network growth property (defined by edge counts) from Shamsi et al. (2024) as the prediction target.

## 6.1 PREDICTION BASELINES

**Persistence forecast.** For our basic baseline model, we employ a naive setting similar to deterministic heuristics techniques, persistence forecast Salcedo-Sanz et al. (2022), for label prediction. In this approach, we use data from the previous and current weeks to predict the next week's property. If we observe an increasing trend in the number of transactions in the current week compared to the previous week, we predict a similar increasing trend for the following week. This simple model is based on the assumption that trends in transaction networks can persist over time.

**Single-network models.** We use four models from literature including HTGN Yang et al. (2021), GCLSTM Chen et al. (2022), EvolveGCN Pareja et al. (2020) and GraphPulse Shamsi et al. (2024) as our baseline single models. We further explain each model in Appendix Section B. We adopt the standard training process for these models over a single dataset and make predictions for the same dataset. We adopt a $70\% - 15\% - 15\%$ split ratio for the train, validation, and test, respectively, for each token network, and during each epoch, the training model processes all snapshots in chronological order. We train every single model for a minimum of $100$ and a maximum of $250$ epochs with a learning rate set to $15 \times 10^{-4}$. We apply early stopping based on the AUC results on the validation set, with patience and tolerance set to $20$ and $5 \times 10^{-2}$, respectively. Specifically, in HTGN training, the node embeddings are reset at the end of every epoch. To address graph-level tasks, we add an extra graph pooling layer as the final layer. This layer, implemented as a Multi-Layer Perceptron (MLP), takes the mean of all node embeddings, concatenating with four snapshot features at the graph level (including the mean of in-degree, the weight of in-degree, out-degree, and weight of out-degree) and then outputs binary classification prediction. We use Binary Cross-Entropy Loss for performance measurement and Adam Kingma & Ba (2015) as the optimization algorithm. It is important to note that the graph pooling layer, performance measurement, and optimization algorithm are also shared by the multi-network model training setup.

## 6.2 MULTI-NETWORK MODEL TRAINING SETUP

While following a similar training approach as in the single model training, we make specific adjustments for the multi-network model training. We set the number of epochs to $300$ with a learning rate of $10^{-4}$ and a train-validation-test chronological split ratio same as single models. Early stopping is applied based on the validation loss with a tolerance of $5 \times 10^{-2}$ and the patience is set to $30$. The best model is selected based on the validation AUC and used to predict the unseen test dataset. We train six multi-network models, each with a different number of networks corresponding to $2^n$ datasets, where $n \in [1, 6]$. We name each multi-network

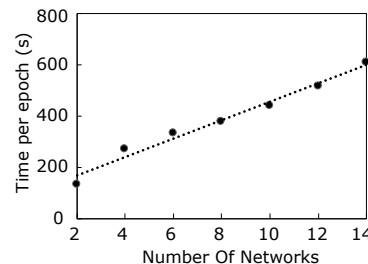

Figure 4: Time per epoch for training multi-network models.

model based on the number of datasets used in training; for example, MN-16 is trained with 16 datasets. For graph property prediction tasks on multi-network, we ran all experiments on NVIDIA Quadro RTX 8000 (48G memory) with 4 standard CPU nodes (either Milan Zen 3 2.8 GHz and 768GB of memory each or Rome Zen 2, 2.5GHz and 256GB of memory each). We repeated each experiment three times and reported the average and standard deviation of different runs. Empirically

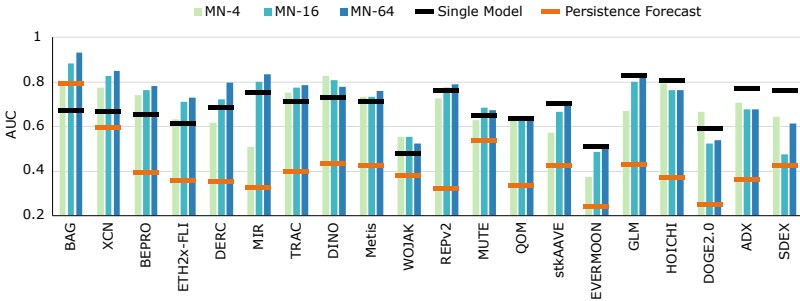

Figure 5: Test AUC of multi-network models trained on 4, 16 and 64 networks and evaluated on unseen test datasets. We compare the performance with persistence forecast, and HTGN models trained and tested on each dataset.

we observe that the TGS-training time scales linearly to the number of networks as seen in Figure 4 where we report the time per epoch for each multi-network model.

## 6.3 RESULTS

**Multi-network vs. single-network models.** We present the performance of our multi-network models trained with datasets of varying sizes and zero-shot inference tested on 20 unseen test datasets. We compare our results with five baseline models: Persistence Forecast, GCLSTM, EvolveGCN, HTGN, and GraphPulse, as explained in Section 6.1. For visual clarity, Figure 5 shows the AUC on test data results for MN-4, MN-16 and MN-64 only as well as persistence forecasting and HTGN single model. We show the performance of all six multi-network models in Appendix Figure 7. Overall, an upward trend is observed in most datasets from multi-network models 2 to 64,

Table 1: Rank-based prediction performance results over different models.

| Model | Top rank ↑ | Avg. rank ↓ | Win ratio ↑ |
|---|---|---|---|
| Persist. forecast | 0 | 7.9 | 0.00 |
| Single model | 3 | 4.35 | - |
| MN-2 | 0 | 6.15 | 0.25 |
| MN-4 | 2 | 4.35 | 0.45 |
| MN-8 | 1 | 4.45 | 0.45 |
| MN-16 | 1 | 3.45 | 0.65 |
| MN-32 | 2 | 3.20 | 0.70 |
| MN-64 | **11** | **2.15** | **0.80** |

such as in BAG, MIR and BEPRO datasets, highlighting the power of larger multi-network models in temporal graph learning. In Figure 5, the MN-64 yields the best AUC in 16 out of 20 test datasets. **This result is significant because the multi-network models outperform the single models specifically trained on these datasets.** We detail the prediction performance of the models in Table 2, where we present the AUC values for both single-trained baselines and multi-network models, specifically MN-32 and MN-64, across various datasets. We also report the Top Rank, Average Rank, and Win Ratio for each model. The Top Rank indicates the number of datasets where a method ranks first. To calculate the Average Rank, we assign an AUC-based rank (ranging from 1 to 8) to every model across the 20 test datasets and compute the average. The Win Ratio represents the proportion of datasets where a model outperforms a single model.

Overall, MN-64 exhibits the best generalization performance, achieving the highest AUC in 6 datasets and second-best in 7 datasets among 20 test datasets in a zero-shot setting. Moreover, Appendix Table 7 indicates that the MN-64 also achieves superior performance in 43 datasets and equivalent performance in 2 datasets among 64 token networks in the training sets compared to the performance of single models. This demonstrates the strong generalizability and transferability of our MN-64 model. While GraphPulse achieves the highest top rank of 8, it relies on trained inference, unlike our multi-network models, which are based on zero-shot inference. Notably, training GraphPulse on each dataset is computationally expensive, while inference testing of our pre-trained MN-64 on all datasets takes only a few minutes. This makes the performance of MN-64, a zero-shot inference model, even more remarkable. Furthermore, despite trained models like HTGN or GCLSTM performing well on certain datasets, our MN-64 model consistently achieves competitive rankings across all datasets. We examine the data selection for different multi-network models and as shown in Section F the performance gain is due to the number of datasets in training and not the bias in data selection.

**Effect of scaling.** In Table 1, we further compare the models by reporting the top rank, average rank, and win ratio for different configurations of the multi-network models. We observe a notable improvement in performance as the number of training networks increases. For instance, the average rank improves from 6.15 for MN-2 to 2.15 for MN-64, which signifies a roughly 50% performance

Table 2: **AUC** scores of multi-network models, single models, and persistence forecasts on test sets across three seeds, including comparisons with state-of-the-art models EvolveGCN, GC-LSTM and GraphPulse. The best performance is shown in bold, and the second best is underlined.

| Method / Dataset | Per. Fore. | HTGN | Trained Inference GCLSTM | EvolveGCN | GraphPulse | Zero-Shot Inference MN-32 | MN-64 |
|---|---|---|---|---|---|---|---|
| WOJAK | 0.378 | $0.479 \pm 0.005$ | $0.484 \pm 0.000$ | $0.505 \pm 0.023$ | $0.467 \pm 0.030$ | $\mathbf{0.534} \pm \mathbf{0.017}$ | $\underline{0.524} \pm 0.027$ |
| DOGE2.0 | 0.250 | $\mathbf{0.590} \pm \mathbf{0.059}$ | $0.538 \pm 0.000$ | $\underline{0.551} \pm 0.022$ | $0.384 \pm 0.180$ | $0.551 \pm 0.022$ | $0.538 \pm 0.038$ |
| EVERMOON | 0.241 | $0.512 \pm 0.023$ | $\mathbf{0.562} \pm \mathbf{0.179}$ | $0.451 \pm 0.046$ | $0.519 \pm 0.130$ | $\underline{0.543} \pm 0.075$ | $0.517 \pm 0.039$ |
| QOM | 0.334 | $0.633 \pm 0.017$ | $0.612 \pm 0.001$ | $0.618 \pm 0.002$ | $\mathbf{0.775} \pm \mathbf{0.011}$ | $\underline{0.669} \pm 0.034$ | $0.647 \pm 0.019$ |
| SDEX | 0.423 | $\mathbf{0.762} \pm \mathbf{0.034}$ | $0.720 \pm 0.002$ | $\underline{0.733} \pm 0.028$ | $0.436 \pm 0.030$ | $0.536 \pm 0.042$ | $0.614 \pm 0.020$ |
| ETH2x-FLI | 0.355 | $0.610 \pm 0.059$ | $0.670 \pm 0.009$ | $0.688 \pm 0.010$ | $0.666 \pm 0.047$ | $\underline{0.715} \pm 0.032$ | $\mathbf{0.729} \pm \mathbf{0.015}$ |
| BEPRO | 0.393 | $0.655 \pm 0.038$ | $0.632 \pm 0.019$ | $0.610 \pm 0.012$ | $\mathbf{0.783} \pm \mathbf{0.003}$ | $0.776 \pm 0.008$ | $\underline{0.782} \pm 0.003$ |
| XCN | 0.592 | $0.668 \pm 0.099$ | $0.306 \pm 0.092$ | $0.512 \pm 0.067$ | $0.821 \pm 0.004$ | $\underline{0.848} \pm 0.000$ | $\mathbf{0.851} \pm \mathbf{0.043}$ |
| BAG | 0.792 | $0.673 \pm 0.227$ | $0.196 \pm 0.179$ | $0.329 \pm 0.040$ | $\mathbf{0.934} \pm \mathbf{0.020}$ | $0.898 \pm 0.075$ | $\underline{0.931} \pm 0.028$ |
| TRAC | 0.400 | $0.712 \pm 0.071$ | $0.748 \pm 0.000$ | $0.748 \pm 0.000$ | $0.767 \pm 0.001$ | $\underline{0.770} \pm 0.007$ | $\mathbf{0.785} \pm \mathbf{0.008}$ |
| DERC | 0.353 | $0.683 \pm 0.013$ | $0.703 \pm 0.022$ | $0.669 \pm 0.009$ | $\underline{0.769} \pm 0.040$ | $0.756 \pm 0.045$ | $\mathbf{0.798} \pm \mathbf{0.027}$ |
| Metis | 0.423 | $0.715 \pm 0.122$ | $0.646 \pm 0.023$ | $0.688 \pm 0.027$ | $\mathbf{0.812} \pm \mathbf{0.011}$ | $0.753 \pm 0.005$ | $\underline{0.760} \pm 0.025$ |
| REPv2 | 0.321 | $0.760 \pm 0.012$ | $0.725 \pm 0.014$ | $0.709 \pm 0.002$ | $\mathbf{0.830} \pm \mathbf{0.001}$ | $0.773 \pm 0.013$ | $\underline{0.789} \pm 0.020$ |
| DINO | 0.431 | $0.730 \pm 0.195$ | $\mathbf{0.874} \pm \mathbf{0.028}$ | $\underline{0.868} \pm 0.029$ | $0.801 \pm 0.020$ | $0.764 \pm 0.048$ | $0.779 \pm 0.113$ |
| HOICHI | 0.374 | $0.807 \pm 0.047$ | $\mathbf{0.857} \pm \mathbf{0.000}$ | $\underline{0.856} \pm 0.001$ | $0.714 \pm 0.010$ | $0.731 \pm 0.029$ | $0.765 \pm 0.018$ |
| MUTE | 0.536 | $0.649 \pm 0.015$ | $0.593 \pm 0.030$ | $0.617 \pm 0.010$ | $\mathbf{0.779} \pm \mathbf{0.004}$ | $0.657 \pm 0.035$ | $\underline{0.673} \pm 0.013$ |
| GLM | 0.427 | $\underline{0.830} \pm 0.029$ | $0.451 \pm 0.003$ | $0.501 \pm 0.033$ | $0.769 \pm 0.018$ | $0.826 \pm 0.035$ | $\mathbf{0.831} \pm \mathbf{0.024}$ |
| MIR | 0.327 | $0.750 \pm 0.005$ | $0.768 \pm 0.026$ | $0.745 \pm 0.015$ | $0.689 \pm 0.097$ | $\underline{0.809} \pm 0.042$ | $\mathbf{0.836} \pm \mathbf{0.016}$ |
| stkAAVE | 0.426 | $0.702 \pm 0.042$ | $0.368 \pm 0.011$ | $0.397 \pm 0.022$ | $\mathbf{0.743} \pm \mathbf{0.006}$ | $0.696 \pm 0.027$ | $\underline{0.709} \pm 0.022$ |
| ADX | 0.362 | $\underline{0.769} \pm 0.018$ | $0.723 \pm 0.002$ | $0.718 \pm 0.004$ | $\mathbf{0.784} \pm \mathbf{0.002}$ | $0.671 \pm 0.015$ | $0.679 \pm 0.024$ |
| Top rank ↑ | 0 | 2 | 3 | 0 | **8** | 1 | $\underline{6}$ |
| Avg. rank ↓ | 6.20 | 3.85 | 4.30 | 4.45 | $\underline{3.00}$ | 3.05 | **2.04** |

enhancement when scaling from two networks to sixty-four. The improvement in the win ratio is also substantial, with MN-64 achieving the highest win ratio of $0.80$, outperforming the other models in most datasets. This indicates that increasing the number of networks in multi-network models significantly enhances their robustness and predictive power, particularly when compared to single models and smaller multi-network configurations.

**Ablation Study** We conducted an ablation study for the TGS-train algorithm to assess the effects of resetting memory (context switching) and shuffling data (IID training). Models are trained same as multi-network

Table 3: Ablation study results (AUC) demonstrating the impact of various training strategies on model performance.

| Model | MN-4 ↑ | MN-8 ↑ | MN-16 ↑ | MN-32 ↑ | MN-64 ↑ |
|---|---|---|---|---|---|
| Base Model | $0.667 \pm 0.111$ | $0.676 \pm 0.099$ | $0.704 \pm 0.115$ | $0.714 \pm 0.107$ | $0.727 \pm 0.114$ |
| w/o IID training | $0.647 \pm 0.113$ | $0.643 \pm 0.117$ | $0.690 \pm 0.105$ | $0.709 \pm 0.093$ | $0.710 \pm 0.123$ |
| w/o Context Switching | $0.667 \pm 0.120$ | $0.608 \pm 0.102$ | $0.693 \pm 0.099$ | $0.713 \pm 0.126$ | $0.664 \pm 0.113$ |

model training setup and tested on the 20 unseen test dataset. The average results are presented in Table 3. Training different multi-network models without resetting memory revealed that persistent memory across epochs negatively impacts generalization, emphasizing the importance of reset mechanisms to reduce overfitting. Additionally, we explored the necessity of shuffling data by fixing the order of training networks. The observed performance decline indicated that incorporating randomness is vital for improving the model's robustness and generalizability.

# 7 CONCLUSION

In this work, we seek to address the question: given a collection of observed temporal graphs, can we predict the evolution of an unseen network within the same domain? We find that it is indeed possible to learn from temporal networks in the same domain and forecast future trends for unseen networks. First, we collected and released a collection of $84$ temporal networks for the temporal graph property prediction task. These datasets serve as the foundation for studying neural scaling laws and foundation models on temporal graphs. Next, to learn from a large number of temporal graphs, we present TGS-train, the first algorithm for training TGNNs across multiple temporal networks. Experimentally, we show that the neural scaling law also applies to temporal graphs; in particular, the more training networks are used, the better the model performance on unseen test networks. In addition, our trained multi-network models can outperform single models trained on individual test networks. Our empirical observations show the high potential of training foundational models on temporal graphs. We believe our TGS method will pave the way for advancements in temporal graph foundation models, providing valuable resources that the community can utilize.

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
