# A    ADDITIONAL DATASET STATISTICS

We summarize detailed statistics of each token network in TGS datasets in Table 4. In the table, the growth rate is the ratio of label 1, indicating the increase in the number of edge counts with respect to the problem definition defined in Section 3. In addition, the novelty score, the average ratio of new edges in each timestamp, and the surprise score, the ratio of edges that only appear in the test set, introduced by Poursafaei et al. Poursafaei et al. (2022), are defined as followed:

$$novelty = \frac{1}{T} \sum_{t=1}^{T} \frac{|E^t \setminus E^t_{seen}|}{|E^t|}, \tag{1a}$$

$$surprise = \frac{|E_{test} \setminus E_{train}|}{|E_{test}|}, \tag{1g}$$

where $E^t$ and $E^t_{seen}$ denotes the set of edges present only in timestamp $t$ and seen in previous timestamps, respectively. $E_{test}$ represents edges that appear in the test set, and edges appearing in the train set are represented as $E_{train}$.

**Comparison between training and testing set**. Nodes, transactions, and length (in days) distribution over the training and testing sets are shown in Figure 6. Training sets well-support the multi-network model to generalize characteristics of the entire TGS dataset due to the similarity between nodes, edge and length in days distributions shown in Figures 6a, 6b, 6c and those distributions across 84 token networks of TGS datasets. In addition, the variance of datasets' characteristics of the testing set is shown in Figures 6d, 6e and 6f.

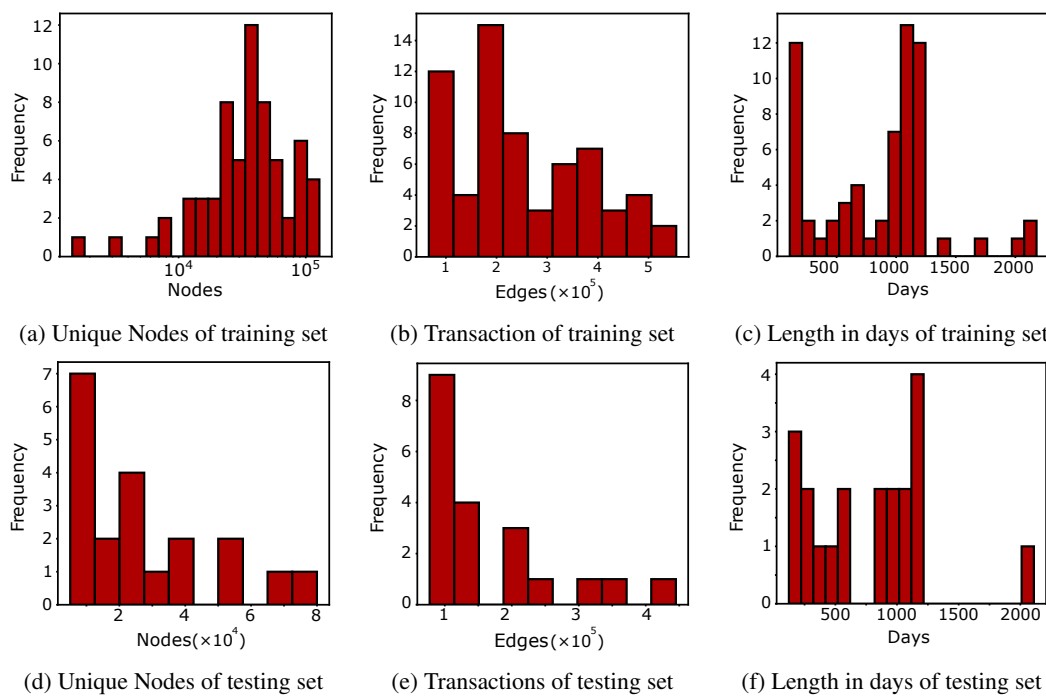

(a) Unique Nodes of training set    (b) Transaction of training set    (c) Length in days of training set

(d) Unique Nodes of testing set    (e) Transactions of testing set    (f) Length in days of testing set

Figure 6: Distribution of the characteristics of the datasets over training and testing sets.

Table 4: All token networks' statistics.

| Token | Node | Transaction | Timestamp (days) | Growth rate | Novelty | Surprise |
|---|---|---|---|---|---|---|
| ARC | 11325 | 70968 | 606 | 0.43 | 0.32 | 0.88 |
| CELR | 65350 | 235807 | 1691 | 0.49 | 0.56 | 0.96 |
| CMT | 86895 | 205961 | 309 | 0.45 | 0.72 | 0.92 |
| DRGN | 113453 | 341849 | 2164 | 0.44 | 0.57 | 0.97 |
| GHST | 35156 | 180955 | 1146 | 0.43 | 0.51 | 0.93 |
| INU | 8556 | 66315 | 154 | 0.27 | 0.41 | 0.59 |
| IOTX | 63079 | 288469 | 1993 | 0.45 | 0.56 | 0.99 |
| QSP | 117977 | 299671 | 2178 | 0.45 | 0.67 | 0.99 |
| REP | 83282 | 224843 | 346 | 0.46 | 0.69 | 0.96 |
| RFD | 23208 | 173695 | 169 | 0.3 | 0.39 | 0.6 |
| TNT | 88247 | 316352 | 1216 | 0.43 | 0.55 | 0.93 |
| TRAC | 71667 | 299181 | 2110 | 0.46 | 0.54 | 0.97 |
| RLB | 28033 | 240291 | 129 | 0.43 | 0.49 | 0.76 |
| steCRV | 19079 | 211538 | 1033 | 0.45 | 0.53 | 0.9 |
| ALBT | 63042 | 434881 | 1152 | 0.43 | 0.44 | 0.89 |
| POLS | 128159 | 554705 | 1132 | 0.45 | 0.61 | 0.94 |
| SWAP | 69230 | 509769 | 1213 | 0.46 | 0.45 | 0.79 |
| SUPER | 83299 | 502030 | 986 | 0.47 | 0.46 | 0.85 |
| RARI | 87186 | 502960 | 1207 | 0.43 | 0.47 | 0.91 |
| KP3R | 39323 | 493258 | 1102 | 0.43 | 0.33 | 0.88 |
| MIR | 79984 | 444998 | 1066 | 0.45 | 0.43 | 0.92 |
| aUSDC | 23742 | 475680 | 1067 | 0.46 | 0.4 | 0.73 |
| LUSD | 25852 | 430473 | 943 | 0.48 | 0.36 | 0.87 |
| PICKLE | 28498 | 430262 | 1149 | 0.48 | 0.34 | 0.69 |
| DODO | 47046 | 390443 | 1131 | 0.47 | 0.45 | 0.91 |
| YFII | 43964 | 391984 | 1196 | 0.44 | 0.44 | 0.96 |
| STARL | 71590 | 369913 | 856 | 0.46 | 0.48 | 0.86 |
| LQTY | 34687 | 374230 | 943 | 0.45 | 0.34 | 0.91 |
| FEG | 118294 | 367584 | 1007 | 0.4 | 0.62 | 0.92 |
| AUDIO | 91218 | 362685 | 1108 | 0.45 | 0.58 | 0.95 |
| OHM | 45728 | 377068 | 690 | 0.43 | 0.46 | 0.88 |
| WOOL | 16874 | 351178 | 716 | 0.41 | 0.18 | 0.41 |
| Metis | 52586 | 343141 | 907 | 0.44 | 0.48 | 0.89 |
| cDAI | 52753 | 358050 | 1437 | 0.45 | 0.46 | 0.9 |
| BITCOIN | 34051 | 347054 | 178 | 0.48 | 0.39 | 0.63 |
| INJ | 60472 | 312822 | 1113 | 0.46 | 0.52 | 0.98 |
| MIM | 23038 | 269366 | 885 | 0.44 | 0.4 | 0.89 |
| GLM | 53385 | 234912 | 1080 | 0.5 | 0.53 | 0.96 |
| Mog | 14590 | 240680 | 107 | 0.37 | 0.38 | 0.55 |
| DPI | 40627 | 234246 | 1150 | 0.49 | 0.5 | 0.86 |
| LINA | 45342 | 227147 | 1144 | 0.45 | 0.46 | 0.95 |
| Yf-DAI | 22466 | 226875 | 1158 | 0.42 | 0.31 | 0.87 |
| BOB | 42806 | 212099 | 199 | 0.35 | 0.48 | 0.73 |
| RGT | 35277 | 211932 | 1110 | 0.44 | 0.46 | 0.98 |
| TVK | 42539 | 208082 | 1062 | 0.41 | 0.48 | 0.93 |
| RSR | 50645 | 205906 | 659 | 0.47 | 0.62 | 0.91 |
| WOJAK | 34341 | 198653 | 201 | 0.37 | 0.48 | 0.73 |
| ANT | 36517 | 200262 | 1107 | 0.47 | 0.46 | 0.93 |
| LADYS | 37486 | 192176 | 181 | 0.37 | 0.52 | 0.79 |
| ETH2x-FLI | 11008 | 199088 | 965 | 0.47 | 0.28 | 0.84 |
| TURBO | 38638 | 189048 | 189 | 0.33 | 0.48 | 0.72 |
| REPv2 | 39061 | 191367 | 1194 | 0.48 | 0.5 | 0.97 |
| NOIA | 29798 | 185528 | 1133 | 0.46 | 0.37 | 0.7 |
| 0x0 | 21531 | 182430 | 283 | 0.51 | 0.46 | 0.81 |
| PSYOP | 25450 | 168896 | 169 | 0.32 | 0.39 | 0.59 |
| ShibDoge | 40023 | 134697 | 680 | 0.43 | 0.53 | 0.8 |
| ADX | 14567 | 123755 | 1188 | 0.44 | 0.4 | 0.91 |
| BAG | 11860 | 122634 | 298 | 0.31 | 0.44 | 0.87 |
| QOM | 21757 | 118292 | 598 | 0.46 | 0.41 | 0.81 |
| BEPRO | 26521 | 120261 | 1132 | 0.46 | 0.48 | 0.87 |
| AIOZ | 29231 | 119926 | 947 | 0.43 | 0.49 | 0.89 |
| PRE | 40476 | 118625 | 1113 | 0.5 | 0.55 | 0.86 |
| CRU | 19990 | 117712 | 1144 | 0.5 | 0.43 | 0.95 |
| POOH | 27245 | 111641 | 193 | 0.26 | 0.49 | 0.69 |
| DERC | 24277 | 111205 | 824 | 0.45 | 0.49 | 0.83 |
| stkAAVE | 37355 | 110924 | 1128 | 0.42 | 0.57 | 0.71 |
| BTRFLY | 8450 | 108371 | 453 | 0.48 | 0.34 | 0.44 |
| SDEX | 9127 | 104869 | 240 | 0.41 | 0.44 | 0.75 |
| XCN | 20085 | 104185 | 607 | 0.46 | 0.42 | 0.84 |
| HOP | 37004 | 102650 | 514 | 0.41 | 0.6 | 0.88 |
| MAHA | 18401 | 96180 | 749 | 0.43 | 0.47 | 0.91 |
| DINO | 15837 | 94140 | 358 | 0.44 | 0.44 | 0.74 |
| bendWETH | 1454 | 96898 | 593 | 0.51 | 0.21 | 0.51 |
| PUSH | 14501 | 93103 | 936 | 0.46 | 0.38 | 0.83 |
| SPONGE | 25852 | 90468 | 184 | 0.31 | 0.66 | 0.81 |
| sILV2 | 12838 | 92905 | 611 | 0.4 | 0.34 | 0.48 |
| SLP | 6675 | 95368 | 1151 | 0.43 | 0.36 | 0.91 |
| crvUSD | 2950 | 88647 | 174 | 0.61 | 0.37 | 0.73 |
| MUTE | 12426 | 82345 | 977 | 0.43 | 0.46 | 0.95 |
| EVERMOON | 7552 | 79868 | 163 | 0.24 | 0.35 | 0.52 |
| HOICHI | 5075 | 77361 | 436 | 0.36 | 0.32 | 0.71 |
| DOGE2.0 | 7664 | 79047 | 123 | 0.45 | 0.38 | 0.66 |
| ORN | 44010 | 239451 | 1134 | 0.46 | 0.47 | 0.87 |
| aDAI | 13648 | 187050 | 1068 | 0.45 | 0.46 | 0.82 |

## B  TEMPORAL GRAPH REPRESENTATION LEARNING METHODS

In this section, we give further details about the temporal graph learning models we used as a baseline for our work.

**HTGN** leverages the power of hyperbolic geometry, which is well-suited for capturing hierarchical structures and complex relationships in temporal networks. HTGN maps the temporal graph into hyperbolic space and utilizes hyperbolic graph neural networks and hyperbolic gated recurrent neural networks to model the evolving dynamics. It incorporates two key modules that are hyperbolic temporal contextual self-attention (HTA) and hyperbolic temporal consistency (HTC)-to ensure that temporal dependencies are effectively captured and that the model is both stable and generalizable across various tasks Yang et al. (2021).

**GraphPulse** addresses the challenge of learning from nodes and edges with different timestamps, which many existing models struggle with. It combines two key techniques: the Mapper method from topological data analysis to extract clustering information from graph nodes and Recurrent Neural Networks (RNNs) for temporal reasoning. This principled approach helps capture both the structure and dynamics of evolving graphs Shamsi et al. (2024).

**GCLSTM** combines a Graph Convolutional Network (GCN) and Long Short-Term Memory (LSTM) units to handle both the structural and temporal aspects of evolving networks. The GCN is used to capture the local structural properties of the network at each snapshot, while the LSTM learns the temporal evolution of these snapshots over time Chen et al. (2022).

**EvolveGCN** is designed to capture the temporal dynamics of graph-structured data. Instead of relying on static node embeddings, EvolveGCN evolves the parameters of a graph convolutional network (GCN) over time. By using a recurrent neural network (RNN) to adapt the GCN parameters, this model is capable of dynamically adjusting during both training and testing, allowing it to handle evolving graphs, even when node sets vary significantly across different time steps Pareja et al. (2020).

## C  TEMPORAL GRAPH PROPERTY PREDICTION

### C.1  NETWORK GROWTH/SHRINK

In this study, we define graph property prediction as the task of predicting a specific graph property. In our case, this involves predicting the growth or shrinkage in the number of transactions in the next snapshot. Specifically, given the current weekly snapshot of a network, the objective is to predict the trend—whether the network will experience growth or shrinkage in transaction volume in the following week. This task has significant applications in the financial domain, as it provides insights into the willingness of investors to engage in a network and whether transaction activity is likely to increase. To ensure consistency, we use the same property prediction setting as GraphPulse (Shamsi et al., 2024), and the formal definition of the graph property is as follows:

**Definition.** We define network growth in terms of edge count as the predicted graph property. Let $\mathcal{G}$ represent a graph, $t$ a specific time, $\delta_1$ and $\delta_2$ time intervals, and $E(t_1, t_n)$ the multi-set of edges between times $t_1$ and $t_n$. The property $P$ is formally expressed as:

$$P(\mathcal{G}, t_1, t_n, \delta_1, \delta_2) = \begin{cases} 1, & \text{if } |E(t_n + \delta_1, t_n + \delta_2)| > |E(t_1, t_n)|, \\ 0, & \text{otherwise.} \end{cases}$$

Setting $n = 7$, $\delta_1 = 1$, and $\delta_2 = 7$, we establish a practical graph property with a 7-day prediction window. This choice is particularly relevant in financial contexts, such as Ethereum asset networks, where it can guide investment decisions, and in social network infrastructure, like Reddit, where it supports maintenance planning.

**Insights for Transaction Networks.** The graph growth/shrink property prediction in financial networks forecasts changes in transaction numbers (edge count), revealing trends in network activity. A growth in edge count indicates increased investor engagement, while a shrinkage suggests reduced activity or market hesitation. This property helps guide investment strategies, resource allocation, and risk management by providing insights into the evolving dynamics of transaction networks.

In temporal graphs, property predictions provide valuable insights into the dynamics and behaviors of evolving networks. While this work focuses on specific properties, numerous other characteristics can also be defined in this domain to highlight the significance of temporal graph property predictions. For instance, properties like the temporal global efficiency, temporal-correlation coefficient, and temporal betweenness centrality offer additional perspectives by capturing unique aspects of a graph's temporal evolution. These examples further clarify the importance of studying temporal graph properties and their relevance to understanding complex network dynamics. Below, we formalize these three additional temporal graph properties and explain their relevance and insights which can be used in future works, particularly for transaction networks.

## C.2 TEMPORAL GLOBAL EFFICIENCY

**Definition.** Temporal global efficiency measures how efficiently information can travel across a temporal graph, considering the dynamic nature of node connections. For a temporal graph $\mathcal{G}_t$ at time $t$, let $d_{ij}(t)$ represent the shortest temporal distance between nodes $i$ and $j$. The global efficiency $E_{global}(t)$ is defined as:

$$E_{global}(t) = \frac{1}{N(N-1)} \sum_{i \neq j \in 1,2,\dots N} \frac{1}{d_{ij}(t)},$$

where $N$ is the total number of nodes in the graph. For disconnected node pairs where no temporal path exists, $d_{ij}(t)$ is set to infinity, and the corresponding term in the sum is considered zero. (Dai et al., 2016)

**Insights for Transaction Networks.** In transaction networks, temporal global efficiency can reveal how effectively transactions propagate through the network. A high-efficiency score indicates well-connected networks with fewer bottlenecks, which may reflect a healthy flow of transactions. Conversely, a low-efficiency score could signal congestion or isolation, impacting investor confidence and transaction throughput.

## C.3 TEMPORAL-CORRELATION COEFFICIENT

**Definition.** Temporal-correlation coefficient $C$ is the measure of the overall average probability for an edge to persist across two consecutive time steps (Nicosia et al., 2013). The temporal-correlation coefficient $C$ of snapshot $t_m$ is defined as follows :

$$C_{t_m} = \frac{1}{N} \sum_{i=1}^{N} \frac{\sum_j a_{ij}(t_m) a_{ij}(t_{m+1})}{\sqrt{[\sum_j a_{ij} t_m][\sum_j a_{ij} t_{m+1}]}}$$

where $a_{ij}$ illustrates an entry in the unweighted adjacency matrix of the graph, and $N$ is the total number of nodes at snapshot $t_m$ (Büttner et al., 2016).

**Insights for Transaction Networks.** Temporal-correlation coefficient can highlight the stability or volatility of transaction patterns over time. A high correlation suggests consistent behaviour across snapshots, which could indicate steady transaction volumes or repeat interactions between participants. A low correlation might point to abrupt changes, such as new market participants, significant events, or shifts in transaction trends.

## C.4 TEMPORAL BETWEENNESS CENTRALITY

**Definition.** Temporal betweenness centrality measures how often a node acts as a bridge along the shortest temporal paths in a graph. For a node $v$, its temporal betweenness centrality of each node $u$ at timestamp $t$:

$$B_u^t = \frac{1}{(N-1)(N-2)} \sum_{j \in \mathcal{V}} \sum_{k \in \mathcal{V}} \frac{U(i,t,j,k)}{|S_{jk}|}$$

defined when $S_{jk} \neq \emptyset$, where the function $U$ return the number of shortest temporal paths include node $u$ from node $j$ to $k$. In the case when $S_{jk} = \emptyset$, we set temporal betweenness centrality of node $u$ to 0 (Tang et al., 2010). The betweenness centrality of each snapshot can be obtained by performing an average of the betweenness centrality of each node for each snapshot.

**Insights for Transaction Networks.** In transaction networks, temporal betweenness centrality identifies key participants that facilitate transactions. Nodes with high centrality act as intermediaries, playing a crucial role in maintaining network connectivity. Understanding such nodes can help detect influential investors, hubs of activity, or potential points of failure.

## D HYPERBOLIC TEMPORAL GRAPH NETWORK (HTGN)

Given feature vectors $X_t^E$ of snapshot $t$ in Euclidean space, an HGNN layer first adopts an exponential map to project Euclidean space vectors to hyperbolic space as follows $X_t^{\mathcal{H}} = exp^c(X_t^E)$, and then performs aggregation and activation similar to GNN but in a hyperbolic manner, $\tilde{X}_t^{\mathcal{H}} = \mathbf{HGNN}(X_t^{\mathcal{H}})$. To prevent recurrent neural networks from only emphasizing the most nearby time and to ensure stability along with generalization of the embedding, HTGN uses temporal contextual attention (HTA) to generalize the lastest $w$ hidden states such that $\tilde{H}_{t-1}^{\mathcal{H}} = \mathbf{HTA}(H_{t-w}; ...; H_{t-1})$ Yang et al. (2021). HGRU takes the outputs from HGNN, $\tilde{X}_t^{\mathcal{H}}$, and the attentive hidden state, $\tilde{H}_{t-1}^{\mathcal{H}}$, from HTA as input to update gates and memory cells and then provides the latest hidden state as the output, $H_t^{\mathcal{H}} = \mathbf{HGRU}(\tilde{X}_t^{\mathcal{H}}, \tilde{H}_{t-1}^{\mathcal{H}})$.

To interpret hyperbolic embeddings, Yang et al. (2021) adopt Poincaré ball model with negative curve $-c$, given $c > 0$, coresponds to the Riemannian manifold $(\mathbb{H}^{n,c}) = \{x \in \mathbb{R}^n : c||x||^2 < 1\}$ is an open n-dimensional ball. Given a Euclidean space vector $x_i^E \in \mathbb{R}^d$, we consider it as a point in tangent space $\mathcal{T}_{x'}\mathbb{H}^{d,c}$ and adopt the exponential map to project it into hyperbolic space :

$$x_i^{\mathcal{H}} = exp_{x'}^c(x_i^E) \tag{2}$$

Resulting in $x_i^{\mathcal{H}} \in \mathbb{H}^{d,c}$, which is then served as input to the HGNN layer as follows Yang et al. (2021):

$$\mathbf{m}_i^{\mathcal{H}} = W \otimes^c \mathbf{x}_i^{\mathcal{H}} \oplus^c \mathbf{b}, \tag{3a}$$

$$\tilde{\mathbf{m}}_i^{\mathcal{H}} = \exp_{\mathbf{x}'}^c(\sum_{j \in \mathcal{N}(i)} \alpha_{ij} \log_{\mathbf{x}'}^c(\mathbf{m}_i^{\mathcal{H}})), \tag{3b}$$

$$\tilde{\mathbf{x}}_i^{\mathcal{H}} = \exp_{\mathbf{x}'}^c(\sigma(\log_{\mathbf{x}'}^c(\tilde{\mathbf{m}}_i^{\mathcal{H}}))). \tag{3c}$$

where $W$, $b$ are learnable parameters and hyperbolic activation function $\sigma$ achieved by applying logarithmic and exponential mapping. HGNN leverages attention-based aggregation by assigning attention score $\alpha_{ij}$ to indicate the importance of neighbour $j$ to node $i$, computed as followed:

$$\alpha_{ij} = softmax_{(j \in \mathcal{N}(i))}(s_{ij}) = \frac{\exp(s_{ij})}{\sum_{j' \in \mathcal{N}_i} \exp(s_{ij'})},$$
$$s_{ij} = \text{LeakReLU}(a^T[\log_0^c(m_i^l) \| \log_0^c(m_j^l)]), \tag{4}$$

where $a$ is trainable vector and $\|$ denotes concatenation operation.

The output of HGNN, $\tilde{X}_t^{\mathcal{H}}$, is then used as input to HGRU along with attentive hidden state $\tilde{H}_{t-1}^{\mathcal{H}}$ obtained by HTA, which generalize $H_{t-1}$ to lastest $w$ snapshots $\{\tilde{H}_{t-w}, ..., H_{t-1}\}$ Yang et al. (2021). Operations behind HGRU are characterized by the following equation Yang et al. (2021):

$$X_t^E = \log_{\mathbf{x}'}^c(\tilde{X}_t^{\mathcal{H}}), \tag{5a}$$

$$H_{t-1}^E = \log_{\mathbf{x}'}^c(\tilde{H}_{t-1}^{\mathcal{H}}), \tag{5b}$$

$$P_t^E = \sigma(W_z X_t^E + U_z H_{t-1}^E) \tag{5c}$$

$$R_t^E = \sigma(W_r X_t^E + U_r H_{t-1}^E), \tag{5d}$$

$$\tilde{H}_t^E = \tanh(W_h X_t^E + U_h(R_t \odot H_{t-1}^E)), \tag{5e}$$

$$H_t^E = (1 - P_t^E) \odot \tilde{H}_t^E + P_t^E \odot H_{t-1}^E, \tag{5f}$$

$$H_t^{\mathcal{H}} = \exp_{\mathbf{x}'}^c(H_t^E). \tag{5g}$$

where $W_z, W_r, W_h, U_z, U_r, U_h$ are the trainable weight matrices, $P_t^E$ is the update gate to control the output and $R_t^E$ is the reset gate to balance the input and memory Yang et al. (2021).

## E  ADDITIONAL RESULTS

Here, we present the test results for the six multi-network models trained on different network sizes, as well as the single model and persistence forecast results. Figure 7 illustrates the AUC of these models on the test set. In most datasets, multi-network models outperform the single model, and in all datasets, they outperform the persistence forecast. We have also compared our model against additional state-of-the-art models, specifically including EvolveGCN Pareja et al. (2020), GC-LSTM Chen et al. (2022) and the only model designed for temporal graph properties prediction, GraphPulse Shamsi et al. (2024) as baselines for the test set. In Table 5 and Table 6 the average and standard deviation of AUC and AP are presented respectively for all models.

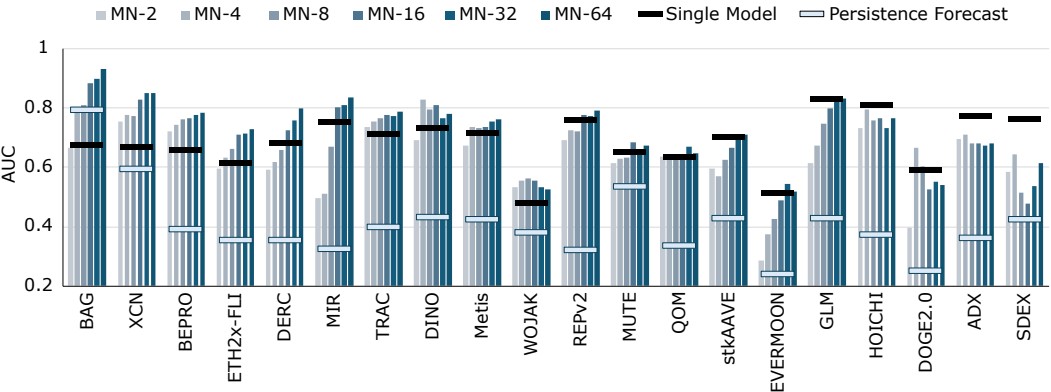

Figure 7: Test AUC of multi-network models trained on $2^n$ datasets where $n \in [1, 6]$ and evaluated on unseen test datasets. Comparing the performance with single models trained and tested on each dataset and persistence forecast results.

## F  EFFECT OF DATA SELECTION ON MULTI-NETWORK MODEL PERFORMANCE

In this section, we investigate the effect of data selection on the performance of multi-network models trained with different training data packs. As the first work on multi-network training for temporal graphs, we explore the importance of our dataset selection process. To avoid any bias, we randomly sampled the training datasets from the 64 available networks. We conducted a novel empirical experiment to examine the impact of dataset selection on training MN models. In this experiment, we choose three disjoint sets of datasets (data pack A, B, and C) for training MN-2, MN-4, MN-8, and MN-16 and two disjoint sets of datasets (data pack A, B) for training MN-32. Using disjoint data packs ensures that each model is trained on unique data, eliminating any overlap that could obscure the results. We then test our models on 20 unseen test datasets.

As shown in Figures 8a the number of training networks increases, the multi-network model performance increases while the variance between different choices of training networks reduces. However,

Table 5: **AUC** scores of multi-network models, single models, and persistence forecasts on test sets across three seeds, including comparisons with state-of-the-art models EvolveGCN, GC-LSTM and GraphPulse. The best performance is shown in bold, and the second best is underlined.

| Token | Per. Fore. | GraphPulse | HTGN | GCLSTM | EvolveGCN | MN-2 | MN-4 | MN-8 | MN-16 | MN-32 | MN-64 |
|---|---|---|---|---|---|---|---|---|---|---|---|
| WOJAK | 0.378 | 0.467 ± 0.030 | 0.479 ± 0.005 | 0.484 ± 0.000 | 0.505 ± 0.023 | 0.534 ± 0.020 | 0.556 ± 0.029 | 0.561 ± 0.018 | 0.556 ± 0.016 | 0.534 ± 0.017 | 0.524 ± 0.027 |
| DOGE2.0 | 0.250 | 0.384 ± 0.18 | 0.590 ± 0.059 | 0.538 ± 0.000 | 0.551 ± 0.022 | 0.397 ± 0.124 | 0.667 ± 0.219 | 0.603 ± 0.080 | 0.526 ± 0.059 | 0.551 ± 0.022 | 0.538 ± 0.038 |
| EVERMOON | 0.241 | 0.519 ± 0.130 | 0.512 ± 0.023 | 0.562 ± 0.179 | 0.451 ± 0.046 | 0.287 ± 0.153 | 0.373 ± 0.037 | 0.426 ± 0.065 | 0.488 ± 0.054 | 0.543 ± 0.075 | 0.517 ± 0.039 |
| QOM | 0.334 | 0.775 ± 0.011 | 0.633 ± 0.017 | 0.612 ± 0.001 | 0.618 ± 0.002 | 0.635 ± 0.061 | 0.624 ± 0.025 | 0.633 ± 0.032 | 0.644 ± 0.009 | 0.669 ± 0.034 | 0.647 ± 0.019 |
| SDEX | 0.423 | 0.436 ± 0.030 | 0.762 ± 0.034 | 0.720 ± 0.002 | 0.733 ± 0.028 | 0.585 ± 0.139 | 0.643 ± 0.021 | 0.515 ± 0.031 | 0.476 ± 0.010 | 0.536 ± 0.042 | 0.614 ± 0.020 |
| ETH2x-FLI | 0.355 | 0.666 ± 0.047 | 0.610 ± 0.059 | 0.670 ± 0.009 | 0.688 ± 0.010 | 0.595 ± 0.083 | 0.632 ± 0.019 | 0.663 ± 0.018 | 0.710 ± 0.037 | 0.715 ± 0.032 | 0.729 ± 0.015 |
| BEPRO | 0.393 | 0.783 ± 0.003 | 0.655 ± 0.038 | 0.632 ± 0.019 | 0.610 ± 0.012 | 0.720 ± 0.028 | 0.742 ± 0.013 | 0.762 ± 0.007 | 0.765 ± 0.024 | 0.776 ± 0.008 | 0.782 ± 0.003 |
| XCN | 0.592 | 0.821 ± 0.004 | 0.668 ± 0.099 | 0.306 ± 0.092 | 0.512 ± 0.067 | 0.754 ± 0.025 | 0.774 ± 0.062 | 0.773 ± 0.076 | 0.827 ± 0.061 | 0.848 ± 0.000 | 0.851 ± 0.043 |
| BAG | 0.792 | 0.934 ± 0.020 | 0.673 ± 0.227 | 0.196 ± 0.179 | 0.329 ± 0.040 | 0.667 ± 0.134 | 0.802 ± 0.155 | 0.808 ± 0.095 | 0.884 ± 0.044 | 0.898 ± 0.075 | 0.931 ± 0.028 |
| TRAC | 0.400 | 0.767 ± 0.001 | 0.712 ± 0.071 | 0.748 ± 0.000 | 0.748 ± 0.000 | 0.734 ± 0.012 | 0.752 ± 0.009 | 0.764 ± 0.012 | 0.776 ± 0.012 | 0.770 ± 0.007 | 0.785 ± 0.008 |
| DERC | 0.353 | 0.769 ± 0.040 | 0.683 ± 0.013 | 0.703 ± 0.022 | 0.669 ± 0.009 | 0.593 ± 0.108 | 0.617 ± 0.030 | 0.657 ± 0.009 | 0.723 ± 0.058 | 0.756 ± 0.045 | 0.798 ± 0.027 |
| Metis | 0.423 | 0.812 ± 0.011 | 0.715 ± 0.122 | 0.646 ± 0.023 | 0.688 ± 0.027 | 0.672 ± 0.045 | 0.734 ± 0.017 | 0.730 ± 0.036 | 0.734 ± 0.016 | 0.753 ± 0.005 | 0.760 ± 0.025 |
| REPv2 | 0.321 | 0.830 ± 0.001 | 0.760 ± 0.012 | 0.725 ± 0.014 | 0.709 ± 0.002 | 0.690 ± 0.024 | 0.725 ± 0.023 | 0.719 ± 0.022 | 0.774 ± 0.013 | 0.773 ± 0.013 | 0.789 ± 0.020 |
| DINO | 0.431 | 0.801 ± 0.020 | 0.730 ± 0.195 | 0.874 ± 0.028 | 0.868 ± 0.029 | 0.692 ± 0.140 | 0.827 ± 0.112 | 0.794 ± 0.096 | 0.809 ± 0.087 | 0.764 ± 0.048 | 0.779 ± 0.113 |
| HOICHI | 0.374 | 0.714 ± 0.010 | 0.807 ± 0.047 | 0.857 ± 0.000 | 0.856 ± 0.001 | 0.733 ± 0.101 | 0.795 ± 0.025 | 0.759 ± 0.040 | 0.763 ± 0.026 | 0.731 ± 0.029 | 0.765 ± 0.018 |
| MUTE | 0.536 | 0.779 ± 0.004 | 0.649 ± 0.015 | 0.593 ± 0.030 | 0.617 ± 0.010 | 0.613 ± 0.027 | 0.627 ± 0.024 | 0.633 ± 0.024 | 0.684 ± 0.042 | 0.657 ± 0.015 | 0.673 ± 0.013 |
| GLM | 0.427 | 0.769 ± 0.018 | 0.830 ± 0.029 | 0.451 ± 0.003 | 0.501 ± 0.033 | 0.613 ± 0.115 | 0.671 ± 0.034 | 0.746 ± 0.082 | 0.800 ± 0.062 | 0.826 ± 0.035 | 0.831 ± 0.024 |
| MIR | 0.327 | 0.689 ± 0.097 | 0.750 ± 0.005 | 0.768 ± 0.026 | 0.745 ± 0.015 | 0.497 ± 0.192 | 0.510 ± 0.015 | 0.669 ± 0.103 | 0.800 ± 0.044 | 0.809 ± 0.022 | 0.836 ± 0.016 |
| stkAAVE | 0.426 | 0.743 ± 0.006 | 0.702 ± 0.042 | 0.368 ± 0.011 | 0.397 ± 0.022 | 0.597 ± 0.076 | 0.571 ± 0.026 | 0.626 ± 0.023 | 0.666 ± 0.033 | 0.696 ± 0.027 | 0.709 ± 0.022 |
| ADX | 0.362 | 0.784 ± 0.002 | 0.769 ± 0.018 | 0.723 ± 0.002 | 0.718 ± 0.004 | 0.695 ± 0.003 | 0.708 ± 0.025 | 0.680 ± 0.008 | 0.678 ± 0.019 | 0.671 ± 0.015 | 0.679 ± 0.024 |

Table 6: **AP** scores of multi-network models, single models, and persistence forecasts on test sets across three seeds, including comparisons with state-of-the-art models EvolveGCN, GC-LSTM and GraphPulse. The best performance is shown in bold, and the second best is underlined.

| Token | Per. Fore. | GraphPulse | HTGN | GCLSTM | EvolveGCN | MN-2 | MN-4 | MN-8 | MN-16 | MN-32 | MN-64 |
|---|---|---|---|---|---|---|---|---|---|---|---|
| WOJAK | 0.658 | 0.863 ± 0.006 | 0.812 ± 0.003 | 0.812 ± 0.000 | 0.827 ± 0.017 | 0.832 ± 0.009 | 0.836 ± 0.015 | 0.842 ± 0.015 | 0.850 ± 0.006 | 0.842 ± 0.004 | 0.837 ± 0.019 |
| DOGE2.0 | 0.2 | 0.966 ± 0.002 | 0.933 ± 0.010 | 0.925 ± 0.000 | 0.927 ± 0.004 | 0.889 ± 0.031 | 0.940 ± 0.050 | 0.936 ± 0.014 | 0.920 ± 0.014 | 0.927 ± 0.004 | 0.921 ± 0.014 |
| EVERMOON | 0.469 | 0.768 ± 0.01 | 0.585 ± 0.065 | 0.612 ± 0.200 | 0.494 ± 0.017 | 0.442 ± 0.059 | 0.508 ± 0.045 | 0.542 ± 0.031 | 0.530 ± 0.040 | 0.567 ± 0.053 | 0.551 ± 0.021 |
| QOM | 0.315 | 0.840 ± 0.002 | 0.623 ± 0.024 | 0.592 ± 0.001 | 0.597 ± 0.002 | 0.632 ± 0.070 | 0.617 ± 0.002 | 0.616 ± 0.007 | 0.626 ± 0.020 | 0.648 ± 0.027 | 0.635 ± 0.027 |
| SDEX | 0.212 | 0.662 ± 0.017 | 0.825 ± 0.048 | 0.725 ± 0.002 | 0.750 ± 0.025 | 0.723 ± 0.039 | 0.725 ± 0.021 | 0.650 ± 0.046 | 0.628 ± 0.036 | 0.697 ± 0.064 | 0.699 ± 0.021 |
| ETH2x-FLI | 0.381 | 0.836 ± 0.015 | 0.590 ± 0.103 | 0.735 ± 0.018 | 0.607 ± 0.122 | 0.621 ± 0.039 | 0.658 ± 0.057 | 0.745 ± 0.051 | 0.737 ± 0.049 | 0.784 ± 0.007 | 0.816 ± 0.014 |
| BEPRO | 0.374 | 0.802 ± 0.001 | 0.686 ± 0.042 | 0.637 ± 0.022 | 0.622 ± 0.009 | 0.743 ± 0.033 | 0.769 ± 0.015 | 0.799 ± 0.016 | 0.804 ± 0.034 | 0.815 ± 0.007 | 0.816 ± 0.014 |
| XCN | 0.413 | 0.793 ± 0.002 | 0.687 ± 0.085 | 0.420 ± 0.032 | 0.555 ± 0.073 | 0.708 ± 0.065 | 0.765 ± 0.080 | 0.781 ± 0.082 | 0.829 ± 0.057 | 0.851 ± 0.023 | 0.861 ± 0.042 |
| BAG | 0.504 | 0.957 ± 0.004 | 0.523 ± 0.290 | 0.235 ± 0.041 | 0.263 ± 0.011 | 0.474 ± 0.152 | 0.699 ± 0.193 | 0.682 ± 0.160 | 0.784 ± 0.118 | 0.829 ± 0.119 | 0.889 ± 0.043 |
| TRAC | 0.4 | 0.767 ± 0.002 | 0.685 ± 0.074 | 0.716 ± 0.006 | 0.722 ± 0.001 | 0.705 ± 0.013 | 0.734 ± 0.012 | 0.741 ± 0.006 | 0.764 ± 0.015 | 0.741 ± 0.015 | 0.758 ± 0.021 |
| DERC | 0.39 | 0.773 ± 0.004 | 0.532 ± 0.021 | 0.621 ± 0.053 | 0.513 ± 0.012 | 0.505 ± 0.157 | 0.477 ± 0.021 | 0.516 ± 0.030 | 0.639 ± 0.118 | 0.700 ± 0.080 | 0.741 ± 0.024 |
| Metis | 0.38 | 0.801 ± 0.003 | 0.601 ± 0.187 | 0.575 ± 0.041 | 0.577 ± 0.006 | 0.532 ± 0.126 | 0.645 ± 0.029 | 0.632 ± 0.056 | 0.611 ± 0.021 | 0.647 ± 0.026 | 0.639 ± 0.077 |
| REPv2 | 0.376 | 0.797 ± 0.003 | 0.758 ± 0.033 | 0.691 ± 0.006 | 0.689 ± 0.001 | 0.610 ± 0.063 | 0.619 ± 0.019 | 0.635 ± 0.042 | 0.705 ± 0.027 | 0.721 ± 0.011 | 0.729 ± 0.011 |
| DINO | 0.480 | 0.871 ± 0.026 | 0.747 ± 0.175 | 0.881 ± 0.029 | 0.875 ± 0.024 | 0.738 ± 0.113 | 0.842 ± 0.102 | 0.793 ± 0.094 | 0.824 ± 0.077 | 0.753 ± 0.030 | 0.765 ± 0.119 |
| HOICHI | 0.602 | 0.623 ± 0.003 | 0.666 ± 0.062 | 0.650 ± 0.049 | 0.658 ± 0.011 | 0.531 ± 0.109 | 0.677 ± 0.049 | 0.605 ± 0.037 | 0.609 ± 0.016 | 0.551 ± 0.045 | 0.594 ± 0.012 |
| MUTE | 0.38 | 0.726 ± 0.002 | 0.615 ± 0.049 | 0.504 ± 0.012 | 0.527 ± 0.015 | 0.579 ± 0.023 | 0.612 ± 0.041 | 0.603 ± 0.058 | 0.675 ± 0.032 | 0.609 ± 0.021 | 0.647 ± 0.048 |
| GLM | 0.387 | 0.712 ± 0.047 | 0.797 ± 0.042 | 0.513 ± 0.001 | 0.529 ± 0.013 | 0.598 ± 0.123 | 0.651 ± 0.031 | 0.709 ± 0.088 | 0.783 ± 0.092 | 0.819 ± 0.035 | 0.838 ± 0.032 |
| MIR | 0.405 | 0.766 ± 0.041 | 0.751 ± 0.003 | 0.765 ± 0.012 | 0.752 ± 0.007 | 0.493 ± 0.212 | 0.442 ± 0.024 | 0.645 ± 0.133 | 0.783 ± 0.064 | 0.799 ± 0.015 | 0.811 ± 0.019 |
| stkAAVE | 0.207 | 0.751 ± 0.005 | 0.750 ± 0.020 | 0.506 ± 0.003 | 0.493 ± 0.009 | 0.662 ± 0.066 | 0.622 ± 0.011 | 0.694 ± 0.021 | 0.730 ± 0.037 | 0.741 ± 0.020 | 0.759 ± 0.019 |
| ADX | 0.372 | 0.765 ± 0.003 | 0.758 ± 0.017 | 0.666 ± 0.002 | 0.661 ± 0.017 | 0.638 ± 0.021 | 0.667 ± 0.040 | 0.632 ± 0.010 | 0.621 ± 0.013 | 0.622 ± 0.018 | 0.628 ± 0.012 |

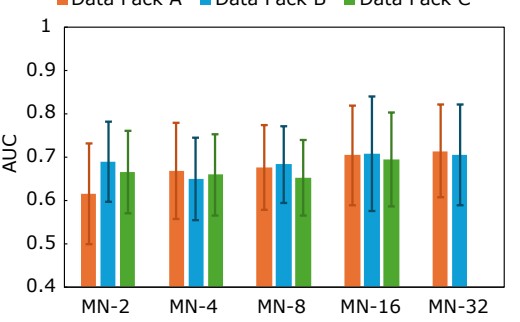

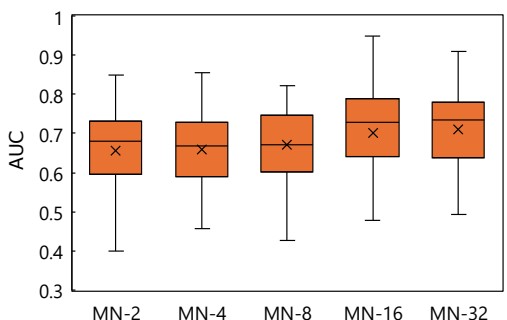

(a) Effect of Data Selection on model performance  (b) Avg. model performance using different data packs

Figure 8: Comparative analysis of data selection effects on model performance.

the difference between models that use the same number of datasets diminishes as we move from models of 2 to 32 datasets. Figure 8b shows the average performance of multi-network models versus the number of training networks used. We observe that smaller models (i.e., MN-2) have a higher variance when compared to larger models (i.e., MN-64); in addition, the model performance also increases from small to large models. For example, MN-64 outperforms MN-32 on 16 out of 20 datasets. While certain datasets, such as ADX, may have a different distribution than other training datasets, overall, we observe that training with more datasets leads to better performance.

Table 7: **AUC** scores of multi-network models, single models, and persistence forecasts on train sets across three seeds. The best performance is shown in bold.

| Token | Per. Fore. | Single Model | MN-64 | Token | Per. Fore. | Single Model | MN-64 |
|---|---|---|---|---|---|---|---|
| POOH | 0.250 | $0.904 \pm 0.008$ | $\mathbf{0.930} \pm \mathbf{0.002}$ | SPONGE | 0.167 | $0.688 \pm 0.032$ | $\mathbf{0.698} \pm \mathbf{0.024}$ |
| MAHA | 0.284 | $0.892 \pm 0.008$ | $\mathbf{0.900} \pm \mathbf{0.001}$ | SWAP | 0.468 | $0.596 \pm 0.044$ | $\mathbf{0.684} \pm \mathbf{0.020}$ |
| PICKLE | 0.321 | $0.841 \pm 0.018$ | $\mathbf{0.877} \pm \mathbf{0.024}$ | MIM | 0.372 | $0.671 \pm 0.014$ | $\mathbf{0.681} \pm \mathbf{0.016}$ |
| TURBO | 0.789 | $0.575 \pm 0.061$ | $\mathbf{0.867} \pm \mathbf{0.008}$ | TVK | 0.376 | $0.460 \pm 0.100$ | $\mathbf{0.679} \pm \mathbf{0.005}$ |
| DODO | 0.346 | $0.739 \pm 0.022$ | $\mathbf{0.851} \pm \mathbf{0.015}$ | OHM | 0.652 | $0.616 \pm 0.008$ | $\mathbf{0.674} \pm \mathbf{0.017}$ |
| KP3R | 0.528 | $0.843 \pm 0.028$ | $\mathbf{0.844} \pm \mathbf{0.027}$ | DRGN | 0.385 | $0.570 \pm 0.067$ | $\mathbf{0.672} \pm \mathbf{0.008}$ |
| Mog | 0.333 | $0.435 \pm 0.042$ | $\mathbf{0.833} \pm \mathbf{0.147}$ | aDAI | 0.434 | $0.521 \pm 0.042$ | $\mathbf{0.668} \pm \mathbf{0.012}$ |
| REP | 0.360 | $0.786 \pm 0.026$ | $\mathbf{0.823} \pm \mathbf{0.063}$ | FEG | 0.442 | $0.484 \pm 0.034$ | $\mathbf{0.601} \pm \mathbf{0.002}$ |
| POLS | 0.393 | $0.708 \pm 0.021$ | $\mathbf{0.822} \pm \mathbf{0.013}$ | STARL | 0.219 | $0.463 \pm 0.028$ | $\mathbf{0.515} \pm \mathbf{0.037}$ |
| AUDIO | 0.441 | $0.802 \pm 0.005$ | $\mathbf{0.821} \pm \mathbf{0.025}$ | crvUSD | 0.291 | $0.367 \pm 0.076$ | $0.367 \pm 0.060$ |
| LINA | 0.428 | $0.773 \pm 0.014$ | $\mathbf{0.814} \pm \mathbf{0.016}$ | RSR | 0.542 | $0.661 \pm 0.075$ | $\mathbf{0.683} \pm \mathbf{0.028}$ |
| ORN | 0.333 | $0.704 \pm 0.018$ | $\mathbf{0.812} \pm \mathbf{0.025}$ | INU | 0.292 | $\mathbf{1.000} \pm \mathbf{0.000}$ | $\mathbf{1.000} \pm \mathbf{0.000}$ |
| SUPER | 0.432 | $0.744 \pm 0.036$ | $\mathbf{0.810} \pm \mathbf{0.002}$ | RLB | 0.273 | $\mathbf{0.981} \pm \mathbf{0.000}$ | $0.846 \pm 0.038$ |
| HOP | 0.415 | $0.284 \pm 0.014$ | $\mathbf{0.810} \pm \mathbf{0.028}$ | sILV2 | 0.581 | $\mathbf{0.887} \pm \mathbf{0.008}$ | $0.857 \pm 0.035$ |
| RARI | 0.440 | $0.753 \pm 0.033$ | $\mathbf{0.809} \pm \mathbf{0.012}$ | PSYOP | 0.403 | $\mathbf{0.863} \pm \mathbf{0.008}$ | $\mathbf{0.863} \pm \mathbf{0.008}$ |
| CRU | 0.431 | $0.719 \pm 0.078$ | $\mathbf{0.808} \pm \mathbf{0.037}$ | RGT | 0.396 | $\mathbf{0.852} \pm \mathbf{0.028}$ | $0.829 \pm 0.009$ |
| ShibDoge | 0.514 | $0.781 \pm 0.042$ | $\mathbf{0.807} \pm \mathbf{0.006}$ | TNT | 0.469 | $\mathbf{0.811} \pm \mathbf{0.046}$ | $0.797 \pm 0.009$ |
| YFII | 0.315 | $0.794 \pm 0.004$ | $\mathbf{0.804} \pm \mathbf{0.018}$ | ARC | 0.532 | $\mathbf{0.800} \pm \mathbf{0.014}$ | $0.746 \pm 0.049$ |
| CELR | 0.495 | $0.729 \pm 0.038$ | $\mathbf{0.788} \pm \mathbf{0.026}$ | CMT | 0.262 | $\mathbf{0.764} \pm \mathbf{0.054}$ | $0.746 \pm 0.016$ |
| LQTY | 0.366 | $0.747 \pm 0.057$ | $\mathbf{0.782} \pm \mathbf{0.010}$ | BOB | 0.105 | $\mathbf{0.748} \pm \mathbf{0.004}$ | $0.623 \pm 0.059$ |
| BITCOIN | 0.382 | $0.544 \pm 0.006$ | $\mathbf{0.782} \pm \mathbf{0.179}$ | PRE | 0.481 | $\mathbf{0.732} \pm \mathbf{0.008}$ | $0.663 \pm 0.013$ |
| AIOZ | 0.390 | $0.745 \pm 0.030$ | $\mathbf{0.769} \pm \mathbf{0.003}$ | IOTX | 0.366 | $\mathbf{0.726} \pm \mathbf{0.020}$ | $0.720 \pm 0.036$ |
| RFD | 0.277 | $0.718 \pm 0.006$ | $\mathbf{0.762} \pm \mathbf{0.023}$ | LUSD | 0.372 | $\mathbf{0.719} \pm \mathbf{0.014}$ | $0.681 \pm 0.022$ |
| ALBT | 0.317 | $0.603 \pm 0.265$ | $\mathbf{0.758} \pm \mathbf{0.009}$ | aUSDC | 0.513 | $\mathbf{0.719} \pm \mathbf{0.019}$ | $0.687 \pm 0.032$ |
| GHST | 0.344 | $0.737 \pm 0.047$ | $\mathbf{0.757} \pm \mathbf{0.005}$ | QSP | 0.431 | $\mathbf{0.693} \pm \mathbf{0.008}$ | $0.680 \pm 0.011$ |
| Yf-DAI | 0.434 | $0.745 \pm 0.008$ | $\mathbf{0.755} \pm \mathbf{0.010}$ | ANT | 0.469 | $\mathbf{0.654} \pm \mathbf{0.064}$ | $0.648 \pm 0.019$ |
| DPI | 0.291 | $0.751 \pm 0.026$ | $\mathbf{0.754} \pm \mathbf{0.012}$ | bendWETH | 0.490 | $\mathbf{0.649} \pm \mathbf{0.039}$ | $0.508 \pm 0.018$ |
| INJ | 0.444 | $0.750 \pm 0.042$ | $\mathbf{0.752} \pm \mathbf{0.066}$ | steCRV | 0.360 | $\mathbf{0.636} \pm \mathbf{0.133}$ | $0.537 \pm 0.016$ |
| LADYS | 0.324 | $0.210 \pm 0.007$ | $\mathbf{0.744} \pm \mathbf{0.022}$ | PUSH | 0.450 | $\mathbf{0.617} \pm \mathbf{0.023}$ | $0.610 \pm 0.052$ |
| cDAI | 0.519 | $0.688 \pm 0.016$ | $\mathbf{0.733} \pm \mathbf{0.022}$ | 0x0 | 0.383 | $\mathbf{0.550} \pm \mathbf{0.021}$ | $0.484 \pm 0.011$ |
| NOIA | 0.359 | $0.616 \pm 0.010$ | $\mathbf{0.719} \pm \mathbf{0.018}$ | SLP | 0.415 | $\mathbf{0.517} \pm \mathbf{0.028}$ | $0.484 \pm 0.002$ |
| WOOL | 0.507 | $0.630 \pm 0.016$ | $\mathbf{0.707} \pm \mathbf{0.125}$ | BTRFLY | 0.127 | $\mathbf{0.851} \pm \mathbf{0.019}$ | $0.763 \pm 0.074$ |

## G  NODE OVERLAP ANALYSIS

We analyze the overlap of nodes between different datasets and within each dataset, which helps demonstrate the highly dynamic nature of our datasets. Specifically, we compared the nodes in each test network with those in the training networks and calculated the average overlap. As shown in Table 8, on average, only 2% of the nodes are common between the training and test datasets, highlighting the rapidly changing structure of these networks. Furthermore, we analyzed the node overlap within each test dataset by splitting it into the standard train-validation-test setup. We compared the nodes in the 70% training snapshots with the nodes in the final 15% test snapshots, and on average, only 4% of the nodes overlapped. This indicates the highly inductive nature of our model and emphasizes the zero-shot challenge it addresses in this domain. These findings underscore the importance of tackling such dynamic and evolving challenges in temporal graph learning.

Table 8: Overlapping Nodes Statistics

| Dataset | Average Node in Common vs Train Set of MN-64 (± std) | Train vs Test Snapshots Node in Common |
|---|---|---|
| MIR | 0.021 ± 0.019 | 0.007 |
| DOGE2.0 | 0.026 ± 0.033 | 0.015 |
| MUTE | 0.033 ± 0.020 | 0.045 |
| EVERMOON | 0.023 ± 0.033 | 0.043 |
| DERC | 0.020 ± 0.020 | 0.031 |
| ADX | 0.024 ± 0.020 | 0.018 |
| HOICHI | 0.023 ± 0.013 | 0.053 |
| SDEX | 0.024 ± 0.019 | 0.141 |
| BAG | 0.019 ± 0.017 | 0.107 |
| XCN | 0.016 ± 0.010 | 0.034 |
| ETH2x-FLI | 0.038 ± 0.041 | 0.028 |
| stkAAVE | 0.026 ± 0.027 | 0.057 |
| GLM | 0.014 ± 0.015 | 0.047 |
| QOM | 0.018 ± 0.014 | 0.044 |
| WOJAK | 0.025 ± 0.032 | 0.018 |
| DINO | 0.018 ± 0.014 | 0.049 |
| Metis | 0.020 ± 0.013 | 0.041 |
| REPv2 | 0.016 ± 0.017 | 0.013 |
| TRAC | 0.015 ± 0.016 | 0.031 |
| BEPRO | 0.023 ± 0.022 | 0.021 |