# OpenReview forum: "Towards Neural Scaling Laws for Foundation Models on Temporal Graphs"
_ICLR.cc/2025/Conference — ICLR 2025 Conference Withdrawn Submission_

### Official Review · Reviewer_FnvK · 2024-11-01

**Soundness:** 1
**Presentation:** 3
**Contribution:** 2
**Rating:** 3
**Confidence:** 4

**Summary:**

This paper studies the scaling laws for foundation models on temporal graphs. It proposes a large number of temporal graphs with various sizes but sharing the same domain. It proposes a training pipeline called TGS-train to train a temporal graph foundation model on a number of temporal graphs and show that the more data is used in training, the better the zero-shot performance of the model on unseen datasets.

**Strengths:**

1. Nice topic.
2. Nice writing and I feel no difficulty understanding most parts.
3. This work can lead to further exploration in graph foundation model, which is a heated ongoing research area.

**Weaknesses:**

1. Unclear technical details and problem statements. Please refer to my questions below.
2. One thing I am concerned about is that scaling law does not only refer to more training data. Bigger model size with deeper networks is also strongly related to the scaling law in my opinion. So for me the title of this paper is somewhat overexaggerated, considering the current focus of this paper is just giving an existing TGNN model with more training data. I think the workload of this paper is not enough for ICLR. The claims made by the authors are intuitive. But for me it is not fully supported with the presented experiments. Please refer to the following points and the questions below.
3. The current baseline models are not enough from my point of view. I would prefer to see the performance of all the models presented in TGB [1] and DyGLib [2].
4. In the datasets proposed in the submission, different graphs share a lot of nodes. This makes me wonder whether we can merge different temporal graphs into a huge one, with edges labeled with the type of tokens. In this sense, the multi newtork assumption just disappears. So I think I cannot get the point of introducing the idea of multi network.
5. And I also think the problem setting in this submission is not zero-shot anymore because by training on the graphs with a set of nodes (addresses), even if we want to predict the behaviour of these nodes related to different tokens, the transaction patterns are decided by the addresses (the users) but not only by the token type. For example, I have a node representing a person and I have seen this person trading on token A and B. Now I learn from this person’s transaction history on A and B and wish to use it to predict the bahavior on token C. We already have prior knowledge of this node and predicting the property on token C becomes not zero-shot anymore. I hope this problem can be well-discussed in rebuttal and I wish the authors can notice me if I have misunderstandings.

[1] Huang, Shenyang, et al. "Temporal graph benchmark for machine learning on temporal graphs." Advances in Neural Information Processing Systems 36 (2024).

[2] Yu, Le, et al. "Towards better dynamic graph learning: New architecture and unified library." Advances in Neural Information Processing Systems 36 (2023): 67686-67700.

**Questions:**

1. Line 153. What do you think of studying CTDGs? I saw you mentioned in DTDGs there are graph dynamics at specific time intervals. But imagine a CTDG with a group of events, each pair of node has its own temporal patterns which can also indicate temporal dynamics. For me, I think DTDGs are discretized CTDGs which will naturally lose part of temporal information, so I think studying CTDGs in your paper scope is more meaningful.
2. Line 169. Could you give definitions of temporal global efficiency, temporal-correlation coefficient, and temporal betweenness centrality? What is temporal graph property prediction? Could you explain by giving examples based on your proposed dataset?
3. Line 282. To be honest, IID training on timeline has been widely used in training temporal knowledge graphs, so I am not sure whether it is that important to point it out in the main body of your paper. I feel it is a slight waste of space.
4. Line 393. I do not understand why you put ablation before your main results. The datasets studied in ablation is omitted so that I do not even know what are you testing with your models. Please notice me if I missed something important.
5. Appendix E is important. I suggest putting it into the main body of the paper. It shows the performance gain is not due to the bias in data selection, but due to the quantity of the training data.
6. The last point I want to mention is that I found nowhere explicitly pointing that the datasets for training your MN series networks are completely disjoint from your evaluation datasets. I guess the “zero-shot” comes from this setting and the MN networks can do inference on the datasets unseen in training. But I prefer that you mention this early in your paper and then the readers can avoid guessing.

---

> ### Author Response · Authors · 2024-11-27
> **Further Clarification from Authors**
>
> Thank you for your valuable feedback which helps us improve the clarity of the paper. We wanted to take this opportunity to clarify our work further. In our study, we trained a temporal model on 64 distinct token networks, each representing a separate financial network with its own set of investors, varying durations, and unique investment behaviors. This is the first study to train a temporal model on such disjoint networks, where each network is independent and exhibits different investment dynamics., With our proposed multi-network training algorithm TGS train, our model is able to effectively capture common structural evolution across these diverse financial networks, demonstrating its ability to generalize and identify shared behaviors.
>
> To evaluate the performance and generalization of our model, we tested it on 20 unseen networks, which share only 2% of their nodes with the training set. This test setup is particularly challenging, as prior works like TGAT, CAWN, and NAT [1,2,3] typically rely on a test set with 10% unseen nodes for inductive testing while our test networks have 98% inductive nodes. The minimal node overlap further underscores the robustness of our model, as it demonstrates the ability to transfer learned knowledge to networks with new investors and behaviors with no fine tuning. By presenting this work, we aim to open a new research direction in temporal graph learning, specifically in multi-network training. Additionally, we will make our datasets available to the research community to encourage further exploration in this area.
>
>
>
> [1] Xu, D., Ruan, C., Korpeoglu, E., Kumar, S., & Achan, K. (2020). Inductive representation learning on temporal graphs. arXiv preprint arXiv:2002.07962.
>
> [2] Wang, Y., Chang, Y. Y., Liu, Y., Leskovec, J., & Li, P. (2021). Inductive representation learning in temporal networks via causal anonymous walks. arXiv preprint arXiv:2101.05974.
>
> [3] Luo, Y., & Li, P. (2022, December). Neighborhood-aware scalable temporal network representation learning. In Learning on Graphs Conference (pp. 1-1). PMLR.

---

> ### Author Response · Authors · 2024-11-27
> **Response by Authors - Part 1**
>
> **W1. Contribution of the paper**
>
> Thank you for raising this point. In this work, we focus on demonstrating the potential of foundation models for temporal graphs. We believe our contributions are significant in several ways:
>
> 1. **Multi-Network Training Algorithm**: While most methods in temporal graph literature are trained and evaluated on a single network, we propose the TGS algorithm, which enables effective multi-network training.
>
> 2. **Transferability Across Networks**: We are the first to show strong transferability across networks, which is a key characteristic of a foundation model and crucial for its generalization.
>
> 3. **Performance Improvement**: Our MN model outperforms single-network models on unseen networks, demonstrating the advantages of multi-network training and the effectiveness of foundation models in temporal graph learning.
>
> By introducing the first multi-network temporal learning approach on a real-world dataset and demonstrating performance improvements over single-model approaches, we aim to open a new research direction toward temporal graph foundation models. This work represents a significant first step in advancing the understanding and application of foundation models in temporal graph learning, and we believe it will contribute substantially to the ongoing development of this emerging field.
>
> **W2. More baseline methods**
>
> We appreciate the reviewer’s feedback regarding more baseline models. It is worth mentioning that our work primarily focuses on graph-level tasks in the Discrete Time Dynamic Graph (DTDG) setting, while models presented in TGB leaderboards and DyGLib are designed for the Continuous Time Dynamic Graph (CTDG) setting for node and link level tasks. Our work also includes GraphPulse [1], the state-of-the-art model for graph property prediction in DTDG, and three other common methods for DTDG.
> To further address your concern, we evaluated another baseline T-GCN [2], as shown in the table below, to make the comparison with more baselines. Without needing any additional training on unseen networks, our MN-64 model outperforms the single-network T-GCN in 18 out of 20 unseen test networks, demonstrating significant transfer capability and highlighting the advantages of our multi-network training approach.
>
> ### Comparison of new baseline’s, T-GCN, performance and MN-64
> | Dataset       | MN-64                | T-GCN                |
> |---------------|----------------------|----------------------|
> | WOJAK         | **0.524 ± 0.027**   | 0.516 ± 0.021        |
> | DOGE2.0       | **0.538 ± 0.038**   | 0.487 ± 0.044        |
> | EVERMOON      | **0.517 ± 0.039**   | 0.463 ± 0.149        |
> | QOM           | **0.647 ± 0.019**   | 0.647 ± 0.032    |
> | SDEX          | 0.614 ± 0.020        | **0.759 ± 0.039**    |
> | ETH2x-FLI     | **0.729 ± 0.015**   | 0.647 ± 0.020        |
> | BEPRO         | **0.782 ± 0.003**   | 0.744 ± 0.074        |
> | XCN           | **0.851 ± 0.043**   | 0.703 ± 0.037        |
> | BAG           | **0.931 ± 0.028**   | 0.334 ± 0.171        |
> | TRAC          | **0.785 ± 0.008**   | 0.741 ± 0.012        |
> | DERC          | **0.798 ± 0.027**   | 0.743 ± 0.077        |
> | Metis         | **0.760 ± 0.025**   | 0.709 ± 0.033        |
> | REPv2         | **0.789 ± 0.020**   | 0.696 ± 0.035        |
> | DINO          | **0.779 ± 0.113**   | 0.544 ± 0.314        |
> | HOICHI        | 0.765 ± 0.018        | **0.836 ± 0.034**    |
> | MUTE          | **0.673 ± 0.013**   | 0.557 ± 0.068        |
> | GLM           | **0.831 ± 0.024**   | 0.531 ± 0.008        |
> | MIR           | **0.836 ± 0.016**   | 0.749 ± 0.026        |
> | stkAAVE       | **0.709 ± 0.022**   | 0.577 ± 0.129        |
> | ADX           | **0.679 ± 0.024**   | 0.674 ± 0.034        |
>
> [1] Shamsi, Kiarash, et al. "GraphPulse: Topological representations for temporal graph property prediction." The Twelfth International Conference on Learning Representations. 2024.
>
> [2] Zhao, Ling, et al. "T-GCN: A temporal graph convolutional network for traffic prediction." IEEE transactions on intelligent transportation systems 21.9 (2019): 3848-3858.

---

> > ### Author Response · Authors · 2024-11-27
> > **Response by Authors - Part 2**
> >
> > **W3. Different graphs share a lot of nodes (Possibility of merge theme into one graph)**
> >
> > Thank you for raising these points. We have conducted a node overlap analysis between the different token networks in our study. As shown in Table 1 in the overall response, the average overlap of nodes between unseen networks and all training networks is approximately 2.21%, which is an insignificant proportion. This indicates that the token networks do not share a significant number of nodes, highlighting the diversity and inductiveness of our contributed networks. These networks are also diverse in several key aspects, including the native types of tokens in transactions, distinct investment patterns, the majority of different investors, as well as varying start dates and durations, which make them separate networks and unsuitable for direct merging. Moreover, merging these networks could reduce the diversity of patterns and limit the effectiveness of the model in capturing the unique characteristics of each token network.
> >
> > The primary focus of our work is to investigate the feasibility of training a multi-network temporal model on different token networks, and to explore the neural scaling law in temporal graphs, a concept well-studied in NLP and CV but less explored in the temporal graph domain. By training on diverse token networks, we aim to shed light on the potential of building a foundation model for temporal graphs. Merging the networks would obscure the unique properties of individual token networks, thus diminishing the clarity and focus of the task.
> >
> >
> > [1] Shamsi, Kiarash, et al. "GraphPulse: Topological representations for temporal graph property prediction." The Twelfth International Conference on Learning Representations. 2024.
> >
> > **W4. Zero-shot setting of this submission clarification**
> >
> > We thank the reviewer for raising this important point. To address this concern, we performed a node overlap analysis to evaluate the proportion of overlapping nodes between each token network in the test set and all token networks in the training set. The results, shown in Table 1 in the overall response, indicate that the average proportion of nodes in each test token network that also appear in the training token networks is 2.21%, which is a trivial proportion. Additionally, we want to clarify that these networks are separate, with their own distinct set of investors, and each network has its own unique characteristics. The properties and behaviors of each network differ, reflecting the diversity across these transaction networks. Moreover, we emphasize that in our setting, we do not use Node IDs as node features, further highlighting the independence and individuality of each network in our study.
> >
> > **W5. What do you think of studying CTDGs?**
> >
> > Thank you for the insightful feedback. In this study, we focus on graph property prediction rather than graph-level tasks to explore the scalability of temporal graphs, as properties offer significant insights into the behavior and evolution of networks. For example, predicting the growth or shrinkage in transaction volume is important in the financial domain, as it helps understand market dynamics and investor behavior. Additionally, predicting graph properties using the DTDGs framework provides valuable insights, particularly in cryptocurrency token networks [1]. Given that DTDGs align well with the objectives of this work, we have chosen to adopt this approach. However, we acknowledge that studying CTDGs presents an interesting research direction, which we intend to pursue in future work. The choice of a 7-day interval is based on the typical weekly activity patterns observed in real-world networks, where behaviors and interactions often follow a cyclical trend. For instance, in blockchain networks, Mondays tend to exhibit the highest activity levels, while weekends often experience a significant drop in transaction volume [2].
> >
> > [1] Akcora, C.G., Li, Y., Gel, Y.R. and Kantarcioglu, M., 2019. Bitcoinheist: Topological data analysis for ransomware detection on the bitcoin blockchain. arXiv preprint arXiv:1906.07852.
> >
> > [2] Abay, N.C., Akcora, C.G., Gel, Y.R., Kantarcioglu, M., Islambekov, U.D., Tian, Y. and Thuraisingham, B., 2019, November. Chainnet: Learning on blockchain graphs with topological features. In 2019 IEEE international conference on data mining (ICDM) (pp. 946-951). IEEE.

---

> > > ### Author Response · Authors · 2024-11-27
> > > **Response by Authors - Part 3**
> > >
> > > **W6. Definitions of temporal global efficiency?**
> > >
> > > Thank you for your valuable feedback. In this study, we focus on **graph property prediction**, specifically predicting the growth or shrinkage of transaction volumes in future weekly snapshots based on the number of transactions in the previous week. Given a current weekly snapshot of a network, our task is to predict whether the network will experience growth or shrinkage in transaction activity in the upcoming week. This task holds significant value in financial networks as it helps forecast investor engagement and potential changes in transaction volume, which can impact investment strategies.
> > >
> > > To address your valuable comment, we have added a Section C to the appendix, where we provide a formal definition of our property prediction setting, as well as additional details. The property prediction we define in this study is based on the growth/shrinkage of the transaction volume, specifically the change in the number of transactions from one week to the next. The formal definition of the graph property prediction is as follows:
> > >
> > > **Graph Property Prediction Definition**
> > >
> > > Let **G** represent a graph, *t* a specific time, and *E(t₁, tₙ)* the multi-set of edges between times *t₁* and *tₙ*. The property *P* is defined as the change in edge count (number of transactions) between consecutive weekly snapshots, as shown:
> > >
> > > P(G, t₁, tₙ, δ₁, δ₂) = 1, if |E(tₙ + δ₁, tₙ + δ₂)| > |E(t₁, tₙ)| 0, otherwise.
> > >
> > >
> > > In this study, we set n = 7, δ₁ = 1, and δ₂ = 7, meaning we predict changes in transaction volume over a 7-day window. This prediction is particularly relevant in the financial domain, where understanding transaction growth or shrinkage can guide investment decisions and provide insights into market dynamics. Additionally, we highlight other potential property predictions in temporal graphs that could be explored in future research. These include temporal global efficiency, temporal-correlation coefficient, and temporal betweenness centrality, among others. We provide the formal definitions and insights on how each of these can be applied to financial networks in Section C of the appendix, where we explain their relevance in the context of transaction networks and how they can offer valuable insights into financial applications. By exploring these property predictions, we aim to provide a broader perspective on temporal graph analysis and encourage further exploration by the research community using our datasets.
> > >
> > > **W7. IID training on timeline has been widely used**
> > >
> > > We appreciate the reviewer’s observation regarding IID training on timelines. While this approach is widely used in the context of temporal knowledge graphs, we included it in the main body to ensure clarity and transparency about our training methodology, particularly for readers who may not be familiar with its application in temporal graph settings. Our approach extends IID training to **multiple temporal networks**, where each network is treated as a separate sequence over time. These sequences, with distinct start and end dates, add complexity compared to standard IID training on a single timeline. This distinction is essential for understanding how our method generalizes and scales across heterogeneous temporal datasets.
> > >
> > >
> > > **W8. Ablation study clarification and reorganization**
> > >
> > > We thank the reviewer for the valuable comment, which has helped in further clarifying the paper. In response to your suggestion, we have moved the ablation study to the results section to improve the paper's consistency and readability. We also add larger MN models for better comparison to our ablation study. This analysis emphasizes the importance of each step in the temporal multi-network training process, which is introduced for the first time in this study. For the datasets, we trained on the same dataset packages used in our multi-network training, scaled to different sizes. Additionally, we tested on 20 unseen networks, and the results were evaluated based on the average AUC across these unseen test networks. This ablation study, which highlights the impact of our method, provides a deeper insight into its effectiveness. The full dataset packages for each set are available on our GitHub repository at [dataset packages](https://anonymous.4open.science/r/ScalingTGNs/data/input/data_list/).
> > >
> > > **W9. Explaining performance gain is not due to the bias in main paper**
> > >
> > > Thank you for the insightful suggestion. We recognized that it is essential to discuss that the performance gain is due to the quantity of training data but not due to the bias in data selection. We added this to the main paper and briefly discussed in the the Result section and reference for more detail to appendix section F.

---

> > > > ### Author Response · Authors · 2024-11-27
> > > > **Response by Authors - Part 4**
> > > >
> > > > **W10. Explicitly pointing disjoint training and evaluation datasets for MN series networks**
> > > >
> > > > We thank you for bringing this to our attention, as it has helped improve the quality of the paper. While we explicitly mentioned this on Line 720 (Appendix section A), we agree that discussing it earlier would enhance the overall readability. In response, we have revised the manuscript to highlight this point more effectively and moved this point to line 322 of the main paper.

---

> ### Comment · Reviewer_FnvK · 2024-11-28
> **Response to Authors**
>
> I really appreciate authors' efforts in explaining details as well as the additional experiments carried out to address my concerns. I checked carefully all of the changes and responses from the authors. Now I am very sure about my judgment and I still hold my view towards a rejection. Sorry for that.
>
> 1. As I mentioned, **in my opinion, scaling law should not be proven by only considering the increasing amount of training data.** I think studying the model size and architecture is also important. In NLP, major SOTA models are Transformer-based and this architecture is empirically proven to be effective. So people are trying to stack more Transformer layers and leveraging more training data which uncovers the scaling law. In this submission, I can only find that the authors are trying to relate to scaling law by introducing more training data on a fixed model structure taken from a baseline model. The choice of the architecture of the baseline model is not perfectly motivated since dynamic graph representation learning methods are diverse in their architectures, not like the role of Transformer in language models.
>
> 2. Another point is that, dynamic graphs can be continuous (CTDGs) or discrete (DTDGs). **If we want to show the scaling law, it is not very comprehensive to only focus on DTDGs.** I think this is also a great concern.
>
> 3. **As pointed out by Reviewer zwwh, I also agree that the transferability across different domains should be considered.** It is a very important characteristic when we refer to scaling law. In NLP, language foundation models are able to do zero-shot inference in the domains that are not frequently observed in the pre-training data (lets call these domains sparse domains here). Of course we cannot expect a strong performance in these sparse domains, but their performance can be improved by scaling the model size as well as the pre-training data. So I believe that it is important to consider cross-domain transferability when we discuss scaling law.
>
> To summarize. I am certain that this work has merits and it is novel, but I think bringing up the notions of "neural scaling laws" and "foundation models" is overexaggerated. Personally, I think the experimental settings and the results cannot lead to constructive conclusions related to neural scaling law on dyanamic graph representation learning. The initial version of the paper was not very consistent in form. After the revision, I can see improvements, but I still believe the storyline of this paper has flaws. So I will keep my score unchanged.

---

### Official Review · Reviewer_1wN3 · 2024-11-02

**Soundness:** 2
**Presentation:** 2
**Contribution:** 2
**Rating:** 3
**Confidence:** 4

**Summary:**

The paper presents a novel study on the transferability and scaling behavior of Temporal Graph Neural Networks (TGNNs) across multiple temporal graphs. The authors introduce the Temporal Graph Scaling (TGS) dataset, comprising 84 ERC20 token transaction networks, and propose the TGS-train algorithm for pre-training TGNNs on multiple networks. The study empirically demonstrates that larger pre-trained models on temporal graphs can achieve enhanced downstream performance on unseen networks, adhering to a neural scaling law. The paper claims this is the first empirical demonstration of transferability to unseen networks in temporal graph learning. The authors also showcase that their multi-network model outperforms fine-tuned TGNNs on thirteen out of twenty unseen test networks using zero-shot inference.

**Strengths:**

1. The introduction of the TGS dataset is a significant contribution to the field, providing a rich resource for researchers to study temporal graph learning and foundation models.
2. The paper provides empirical evidence that the neural scaling law, previously observed in NLP and CV, also applies to temporal graph learning, which is a valuable insight for model training and generalization.
3. The demonstration of transferability of pre-trained models to unseen networks is a crucial step towards developing foundation models for temporal graphs, with potential applications in various domains.

**Weaknesses:**

1. The novelty of the paper is limited to meet the standard of the ICLR. From the data perspective, the authors construct Ethereum and ERC20 Token Networks. In my opinion, it is not comprehensive enough. For example, temporal datasets like DBLP and Stack Overflow in the paper "WinGNN: Dynamic Graph Neural Networks with Random Gradient Aggregation Window". From the method perspective, there seems no architecture or algorithm for modeling.
2. The writing is not clear, e.g., Temporal Graph Property Prediction in preliminaries and construction of Ethereum and ERC20 Token Networks. In methodology, equations are needed to clarify.
3. The effect of scaling needs to be further explored. Table 2 is not enough to verify this strong standpoint.

**Questions:**

1. What is temporal global efficiency, temporal-correlation coefficient, and temporal betweenness centrality in Line 169 of page 4?
2. Figure 2 is not very clear, how do you conduct token parse?

---

> ### Author Response · Authors · 2024-11-27
> **Response by Authors - Part 1**
>
> **W1. Novelty of the paper**
>
> Thanks for your comment. Our main contribution is to demonstrate the **first** successful transfer of tasks across temporal graphs: our multi network model achieves state-of-the-art performance on unseen test networks without any fine tuning, outperforming models that were trained specifically on these test networks.
> Our work enables the study of scaling laws and the application of foundation models across diverse temporal graphs, opening up new possibilities for generalized learning across different networks. This highlights the powerful transferability and scalability of our model and in learning the evolution of temporal graphs and applying this knowledge to new, unseen networks.
> In this sense, our work presents an important advancement in the field of temporal graph learning by introducing the first multi-network model for temporal graph learning. Unlike existing approaches, which focus primarily on static graphs or training on single temporal networks, we propose a novel multi-network training scheme.  Additionally, our work provides the largest collection of temporal graphs, accompanied by tools that enable the construction of foundation models for temporal graph learning. We believe these contributions are highly significant for ICLR as they not only pave the way for future research on temporal graph foundation models but also offer a valuable resource to the research community.
>
> **W2. Clarification of graph property prediction and definition of temporal global efficiency, temporal-correlation coefficient, and temporal betweenness centrality**
>
> Thank you for your insightful feedback. In this work, we focus on temporal graph property prediction which aims at  forecasting properties about the temporal graph for example the growth or contraction of transaction volumes in future weekly snapshots, utilizing the number of transactions from the prior week. Our objective is to determine if a network will exhibit growth or shrinkage in transaction activity during the upcoming week based on the current weekly snapshot. This analysis is particularly crucial in financial networks, as it aids in predicting investor engagement and potential fluctuations in transaction volume, subsequently influencing investment strategies.
>
> In response to your constructive comment, we have added Section C in the appendix, which presents a formal definition of our property prediction framework along with supplementary details. The property prediction established in this study pertains to the growth or shrinkage of transaction volumes, specifically focusing on the alteration in the number of transactions from one week to the next. The formal definition of graph property prediction is articulated as follows:
>
> **Graph Property Prediction Definition**
>
> Let **G** represent a graph, *t* a specific time, and *E(t₁, tₙ)* the multi-set of edges between times *t₁* and *tₙ*. The property *P* is defined as the change in edge count (number of transactions) between consecutive weekly snapshots, as shown:
>
> P(G, t₁, tₙ, δ₁, δ₂) = 1, if |E(tₙ + δ₁, tₙ + δ₂)| > |E(t₁, tₙ)| 0, otherwise.
>
> In this study, we set n = 7, δ₁ = 1, and δ₂ = 7, which implies that we are predicting transaction volume changes over a 7-day span. This forecasting is especially pertinent in the financial sector, where understanding transaction growth or contraction can inform investment choices and improve comprehension of market dynamics. Additionally, we identify several other potential property predictions within temporal graphs that warrant exploration in future research. These include temporal global efficiency, temporal-correlation coefficient, and temporal betweenness centrality, among others.
>
> In Section C of the appendix, we delineate formal definitions and discuss the applicability of each to financial networks, illustrating their relevance to transaction networks and the insights they may provide for financial applications. By investigating these property predictions, we aim to enhance the understanding of temporal graph analysis and motivate further investigation by the research community utilizing our datasets.

---

> > ### Author Response · Authors · 2024-11-27
> > **Response by Authors - Part 2**
> >
> > **W3. The effect of scaling needs to be further explored**
> >
> > Thank you for your insightful comment. In this work, we introduced the first multi-network temporal model trained on disjoint temporal networks. Our focus is to demonstrate the efficiency of this critical task, showing that incorporating temporal data from diverse networks, even with distinct characteristics, significantly enhances model performance. By leveraging data from different temporal graphs, we highlight the potential of scalable and generalized models capable of learning across varied and dynamic domains—a capability that has not been explored previously. This work paves the way for the research community to delve deeper into temporal graph foundation models, setting a foundation for future studies to build upon. By showcasing the effectiveness of a multi-network approach on real-world datasets, we not only achieve superior performance compared to single-network methods but also open a new avenue for research in this emerging domain. As a significant first step, our contribution underscores the promises of scalability and generalization in temporal graph learning, providing a roadmap for advancing this field.
> >
> >
> > **W4. Token parsing clarification**
> >
> > Thank you for your feedback, which has greatly helped us clarify our framework. To create our transaction network data, we start by setting up an Ethereum node and connecting to the peer-to-peer (P2P) network using the Ethereum client, Geth (https://github.com/ethereum/go-ethereum). Once the node is running, we use Ethereum-ETL (https://github.com/blockchain-etl/ethereum-etl) to extract data related to all ERC20 tokens. Ethereum-ETL allows us to gather detailed transaction records between participants, which we use to build token networks representing interactions between different addresses based on ERC20 token transfers. These token networks are then analyzed to understand the flow of assets and relationships between addresses over time. After extracting these token transaction networks, we selected 84 networks and split them into 64 networks for training and 20 unseen networks for testing. These unseen networks are not involved during the training phase. Appendix F illustrates the impact of our data selection process on performance, particularly highlighting the challenges of larger multi-network training.  For each token, we extract its transaction network and create weekly snapshots to track changes over time. Labels are generated for each network to facilitate detailed analysis. In response to your valuable feedback, we have added Section C in the appendix to provide a more thorough explanation of our property prediction task. It is also important to emphasize that each token network is analyzed within its own specific duration, defined by its start and end dates. By accounting for the varying time frames of each token, we ensure that our analysis captures the dynamic and temporal characteristics of transaction networks. We will update the paper with a more detailed figure addressing your concern for the final revised version.

---

### Official Review · Reviewer_zwwh · 2024-11-04

**Soundness:** 2
**Presentation:** 3
**Contribution:** 2
**Rating:** 5
**Confidence:** 4

**Summary:**

This paper focuses on a problem that is to estimate the evolution of an unseen network/graph given some observed temporal graph in the same domain. In this paper, a new data that includes 84 temporal networks/graphs is built for temporal graph property prediction task. For learning from this data, a new method TGS-train is proposed which is used to train a temporal GNN across multiple temporal graphs. The whole idea is clear. Most of GNN approaches on multiple graphs are used on drug discovery, while the temporal GNN methods focus on a single graph. This paper combines these two directions and proposed a reasonable data and method. And this paper explored the graph scaling laws on temporal graph learning which is very useful. It is reasonable that training the TGNN using more temporal graphs will benefit the performance.

**Strengths:**

Strengths:

1.	This paper presents a temporal graph scaling dataset that includes 84 ERC20 token transaction networks. All further analysis about the scaling behavior, and transferability is based on this data.

2.	This paper presents a new training method TGS-training to enable the temporal graph network training on multiple graphs.

**Weaknesses:**

Weaknesses:

1.	Although this paper introduces the transferability of the proposed method, this method is hard to be extended on the graph in a different domain. The general transferability is still limited.

2.	The training method is based on HTGN. The TGNN architecture is not proposed by this paper. The key technical modifications are the shuffling and resets. The technical contribution of these two mechanisms is very limited. Shuffling the training temporal graphs is a very straightforward method. It is not clear what the main architecture and method differences between the proposed method with the HTGN are.

**Questions:**

The proposed training method is very similar to HTGN. Beside shuffling and resets, do we have any other architecture upgrading or modification?

---

> ### Author Response · Authors · 2024-11-27
> **Response from Authors**
>
> **W1.General transferability on different domain**
>
> Thank you for your valuable comment. To the best of our knowledge, this is the first study to demonstrate the transferability across temporal networks within the same domain, establishing a promising direction for future research. To better clarify the different structures of our datasets we provide [Figure](https://anonymous.4open.science/r/ScalingTGNs/pic/data_diversity_high_res.png)  which highlights the substantial diversity in graph structures and distributions in our datasets. We also compared our dataset diversity with the available dataset in [1] which further illustrates how our proposed TGS datasets improved upon the available ones. While we acknowledge the importance of generalizing across multiple domains, we believe the first step is to prove that such transferability is possible within a single domain and provide datasets for the research community to explore this important topic. We agree that learning from multiple domains is an interesting next step which we leave as future work.
>
> [1] Shamsi, K., Poursafaei, F., Huang, S., Ngo, B.T.G., Coskunuzer, B. and Akcora, C.G., 2024, March. GraphPulse: Topological representations for temporal graph property prediction. In The Twelfth International Conference on Learning Representations.
>
>
> **W2. The training method is based on HTGN**
>
> This is different from most TGL work where a single network is trained and tested on, testing completely on unseen network is new and the setting of multi-network training is also new.
>
> Generalization to unseen temporal networks, we assume all networks come from same IID distribution, same distribution for train and test.
>
> We appreciate your valuable feedback. To address the concern about the generalizability and robustness of our results across various GNN architectures, we have included an additional model in our evaluation: GC-LSTM. This model combines graph convolutional networks with long short-term memory units, enabling it to effectively capture both spatial and temporal dependencies in graph data. It serves as an important benchmark for testing the temporal robustness of our findings. Both the GC-LSTM and our HTGN-based model were evaluated under identical conditions, and the results are now presented in the revised manuscript and  [ Figure](https://anonymous.4open.science/r/ScalingTGNs/pic/GCLSTM-SL.pdf) of the rebuttal PDF. While GC-LSTM demonstrates some scaling behavior, our HTGN-based model outperforms it, exhibiting superior scalability across networks.

---

### Official Review · Reviewer_fHiS · 2024-11-08

**Soundness:** 3
**Presentation:** 3
**Contribution:** 3
**Rating:** 6
**Confidence:** 3

**Summary:**

The paper introduces the Temporal Graph Scaling (TGS) benchmark, containing 84 temporal graphs derived from Ethereum transaction networks. Besides, the paper investigated the training scaling potential for temporal graph neural networks (TGNNs), proposes the first algorithm for pre-training TGNNs on multiple temporal graphs and study how transferrable a pre-trained temporal graph model is to unseen networks. The work demonstrates that by pre-training on a substantial set of temporal graphs, the multi-network model can be directly applied to unseen token networks, delivering better performance compared to individual models trained specifically on those test networks.

**Strengths:**

- New Benchmark for Temporal Graph Scaling: This work introduces a new benchmark specifically designed for examining temporal graph scaling, featuring an extensive collection of networks. This benchmark enhances the ability to evaluate temporal graph scaling low across a diverse set of temporal graphs.

- A Multi-Graph Training Algorithm: The authors propose a novel multi-graph training algorithm that shows strong effectiveness on the new benchmark. This approach advances the potential for generalizing training across multiple temporal graphs.

- Promising Empirical Results: The study demonstrates the potential of pre-training temporal graph neural networks (TGNNs) across temporal graphs, highlighting the great potential of pre-training strategies for TGNNs.

**Weaknesses:**

- Limited Ablation Study on Scaling: The ablation study for the TGS-train algorithm—specifically in terms of memory resetting and data shuffling—was only conducted on MN-4 and MN-8. Given the paper’s focus on scaling laws in TGNN pre-training, extending this study to larger networks, such as MN-32 and MN-64, would provide a more comprehensive understanding of scaling effects.

- Restricted Evaluation on Prediction Tasks: The authors use network growth, defined by edge counts, as the prediction target. However, other common tasks for temporal graphs, such as node classification and link prediction, are not included. Including these tasks could enhance the relevance of the benchmark for a wider range of temporal graph applications.

**Questions:**

Clarification on Technical Details in Section 5: Some implementation details are unclear. For example, has a random historical embedding been used, and how exactly does this mitigate catastrophic forgetting? A more detailed explanation on this technique and its benefits would be helpful.

---

> ### Author Response · Authors · 2024-11-27
> **Response from Authors - Part 1**
>
> **W1. Ablation study with larger MN**
>
> We thank you for this suggestion. We have added ablation study on MN-16, MN-32 and MN-64. We observe that the conclusion remains the same: both IID training and Context Switching are essential parts of the TGS training algorithm and each provides a performance boost when applied.
>
>
> | Model                         | MN-4          | MN-8          | MN-16         | MN-32         | MN-64         |
> |-------------------------------|---------------|---------------|---------------|---------------|---------------|
> |        Base Model            | 0.667 ± 0.111 | 0.676 ± 0.099 | 0.704 ± 0.115 | 0.714 ± 0.107 | 0.727 ± 0.114 |
> |        w/o IID training      | 0.647 ± 0.113 | 0.643 ± 0.117 | 0.690 ± 0.105 | 0.709 ± 0.093 | 0.710 ± 0.121 |
> |        w/o Context Switching | 0.667 ± 0.120 | 0.608 ± 0.102 | 0.693 ± 0.099 | 0.713 ± 0.126 | 0.664 ± 0.113 |
>
>
> **W2. Evaluation on other prediction tasks**
>
> Thank you for raising this point. To the best of our knowledge, we are the first to investigate neural scaling law on temporal graphs. Therefore, we would like to focus on a task that is defined over the overall graph structure and potentially transferable across graphs thus choosing the graph property prediction task.
>
> **A significant portion of new nodes is introduced at each snapshot**. In the following table, we compare the number of new nodes coming into the network between pairs of consecutive snapshots, represented as (t, t+1), (t+1, t+2), and so on. The **first column** shows the average percentage of new nodes introduced between each snapshot pair, while the **second column** reports the highest percentage of new nodes observed between any two consecutive snapshots.
> On average, we observe that approximately **20% of nodes are new** with each snapshot, demonstrating the dynamic and evolving nature of the network. Some datasets show a much higher percentage of new nodes, even exceeding **90%**, which is common in financial networks as investors frequently enter and exit the network. This high turnover of nodes reflects the changing nature of participants in financial systems, where new investors may join and others may leave.  This high turnover of nodes makes it challenging for models to effectively learn node and link-level tasks, as these tasks rely on a relatively consistent set of nodes across snapshots.
>
> However, graph-level tasks are more well-defined and offer significant advantages in financial network applications. In particular, predicting graph properties can provide deeper insights into the structural evolution of networks, which is crucial for identifying investment signals in practice. Given this, our work focuses on graph-level tasks to leverage the dynamic nature of financial networks and deliver actionable insights for real-world applications.
>
>
> | Dataset     | Pairwise Snapshot Average New Nodes (%) | Highest Pairwise Snapshot New Nodes (%) |
> |-------------|-------------------------------------|--------------------------------------|
> | MIR         | 20.2                                | 47.9                                 |
> | DOGE2.0     | 19.3                                | 40.8                                 |
> | MUTE        | 21.1                                | 48.9                                 |
> | EVERMOON    | 19.0                                | 30.5                                 |
> | DERC        | 21.4                                | 61.4                                 |
> | ADX         | 19.6                                | 56.4                                 |
> | HOICHI      | 19.1                                | 36.2                                 |
> | SDEX        | 19.6                                | 31.1                                 |
> | BAG         | 21.0                                | 49.1                                 |
> | XCN         | 19.8                                | 96.0                                 |
> | ETH2x-FLI   | 18.7                                | 52.1                                 |
> | stkAAVE     | 23.5                                | 70.5                                 |
> | GLM         | 21.2                                | 80.7                                 |
> | QOM         | 20.2                                | 51.9                                 |
> | WOJAK       | 21.4                                | 34.8                                 |
> | DINO        | 20.1                                | 43.5                                 |
> | Metis       | 21.1                                | 81.7                                 |
> | REPv2       | 20.4                                | 70.0                                 |
> | TRAC        | 21.9                                | 64.7                                 |
> | BEPRO       | 21.4                                | 92.9                                 |

---

> > ### Author Response · Authors · 2024-11-27
> > **Response from Authors - Part 2**
> >
> > **W3. Clarification on technical details in section 5**
> >
> > Thank you for raising this point. In this work, the main goal of our multi-network model is to predict unseen test networks. During training to mitigate catastrophic forgetting, we train with each network sequentially in each epoch. Specifically when switching to a new network, the model adapts to the new network by resetting its node embedding (to zero-embeddings). As more data is received of this network, the node embedding evolves over time and provides predictions for the graph property task.  In this procedure, a single network is tested at a time and then switched to a different one to test. So far we haven’t observed catastrophic forgetting in our setting.

---

### Author Response · Authors · 2024-11-27
**Rebuttal by Authors - Part 1**

We thank the reviewers for their insightful reviews and valuable feedback on our work. We are glad to see that reviewers pointed out that our TGS benchmark “is a significant contribution to the field” (Reviewer 1wN3) and “features an extensive collection of networks” (Reviewer fHiS). our work “explored the graph scaling laws on temporal graph learning which is very useful” (Reviewer zwwh). Lastly, “the demonstration of transferability of pre-trained models to unseen networks is a crucial step towards developing foundation models for temporal graphs” (Reviewer 1wN3).
Here, we would like to further clarify and discuss the contributions and novelty of our work.


**1. Novelty: Why Our Work Is Significant in Temporal Graph**

We are the first to study foundation models on temporal graphs, a domain that has previously been limited to static graph analysis. Existing work has primarily focused on training and testing models on a single temporal graph. In contrast, our approach introduces a multi-network training scheme, allowing for the exploration of scaling laws and the use of foundation models across different temporal networks. This novel framework enables the analysis of how well a model trained on multiple networks can generalize to unseen data.
Moreover, we are the first to demonstrate a significant positive transfer of tasks across temporal graphs. Specifically, our foundation model outperforms models that were trained directly on 10 out of 20 unseen test networks, showcasing its superior generalization ability. This positive transfer highlights the potential for foundation models to learn the evolution of temporal graphs and to effectively transfer this knowledge to downstream networks.
Additionally, our work provides the largest collection of temporal graphs, both in terms of the number of graphs and the diversity of the networks represented. We have also developed the necessary tools for building foundation models on temporal graphs, which will be valuable to the research community. Given these contributions, we believe our work represents a critical and pioneering step toward the development of foundation models for temporal graphs.

---

> ### Author Response · Authors · 2024-11-27
> **Rebuttal by Authors - Part 2**
>
> **2. Node Overlap Analysis**
>
> Based on the comments of the reviewers, we have conducted a computation to highlight the overlap of new nodes in our training and test sets showing that there is minimal overlap between training and test nodes.
>
> We analyze the overlap of nodes between different datasets and within each dataset, which helps demonstrate the highly dynamic nature of our datasets. Specifically, we compared the nodes in each test network with those in the training networks and calculated the average overlap. The results showed that, on average, only 2% of the nodes are common between the training and test networks, highlighting the rapidly changing structure of these networks.
> Furthermore, we analyzed the node overlap within each test dataset by splitting it into the standard train-validation-test setup. We compared the nodes in the 70% training snapshots with the nodes in the final 15% test snapshots, and on average, only 4% of the nodes overlapped. This indicates the highly inductive nature of our model and emphasizes the zero-shot challenge it addresses in this domain. These findings underscore the importance of tackling such dynamic and evolving challenges in temporal graph learning. We have also included this information into the appendix section G of the paper
>
> ## Table 1: Overlapping Nodes Statistics
> | Dataset   | Average Node in Common vs Train Set of MN-64 (± std) | Train vs Test Snapshots Node in Common|
> |-----------|------------------------------------------------------|-------------------------------------|
> | MIR       | 0.021 ± 0.019                                        | 0.007                               |
> | DOGE2.0   | 0.026 ± 0.033                                        | 0.015                               |
> | MUTE      | 0.033 ± 0.020                                        | 0.045                               |
> | EVERMOON  | 0.023 ± 0.033                                        | 0.043                               |
> | DERC      | 0.020 ± 0.020                                        | 0.031                               |
> | ADX       | 0.024 ± 0.020                                        | 0.018                               |
> | HOICHI    | 0.023 ± 0.013                                        | 0.053                               |
> | SDEX      | 0.024 ± 0.019                                        | 0.141                               |
> | BAG       | 0.019 ± 0.017                                        | 0.107                               |
> | XCN       | 0.016 ± 0.010                                        | 0.034                               |
> | ETH2x-FLI | 0.038 ± 0.041                                        | 0.028                               |
> | stkAAVE   | 0.026 ± 0.027                                        | 0.057                               |
> | GLM       | 0.014 ± 0.015                                        | 0.047                               |
> | QOM       | 0.018 ± 0.014                                        | 0.044                               |
> | WOJAK     | 0.025 ± 0.032                                        | 0.018                               |
> | DINO      | 0.018 ± 0.014                                        | 0.049                               |
> | Metis     | 0.020 ± 0.013                                        | 0.041                               |
> | REPv2     | 0.016 ± 0.017                                        | 0.013                               |
> | TRAC      | 0.015 ± 0.016                                        | 0.031                               |
> | BEPRO     | 0.023 ± 0.022                                        | 0.021                               |

---

### Author Response · Authors · 2024-12-03

We would like to express our sincere gratitude to all the reviewers for their constructive and insightful comments. Your valuable feedback has provided us with significant guidance, and we greatly appreciate the time and effort you have dedicated to reviewing our work. We will carefully revise our paper and incorporate your suggestions to enhance the clarity, quality, and impact of the next version. Thank you once again for helping us improve our research.

---

### Note · Authors · 2024-12-03

I have read and agree with the venue's withdrawal policy on behalf of myself and my co-authors.